# EvA: Evolutionary Attacks on Graphs

## Abstract

Even a slight perturbation in the graph structure can cause a significant drop in the accuracy of graph neural networks (GNNs). Most existing attacks leverage gradient information to perturb edges. This relaxes the attack's optimization problem from a discrete to a continuous space, resulting in solutions far from optimal. It also restricts the adaptability of the attack to non-differentiable objectives. Instead, we propose an evolutionary-based algorithm to solve the discrete optimization problem directly. Our Evolutionary Attack (EvA) works with any black-box model and objective, eliminating the need for a differentiable proxy loss. This permits us to design two novel attacks that: reduce the effectiveness of robustness certificates and break conformal sets. We introduce a sparse encoding that results in memory complexity that is linear in the attack budget. EvA reduces the accuracy by an additional $\sim 11\%$ on average compared to the best previous attack, revealing significant untapped potential in designing attacks.

## 1 Introduction

Given the widespread applications of graph neural networks (GNNs), studying their robustness to natural and adversarial noise is of great importance. In node classification, GNNs leverage the edge structure between data points to improve their performance. However, a small perturbation in the graph structure (adding or removing a few edges) can significantly reduce GNNs' accuracy, even below the performance of an MLP (which completely discards the structure). Similar to images and continuous data, most of the proposed (structure) attacks are gradient-based. They compute the gradients of a loss w.r.t. the adjacency matrix and apply a perturbation according to that. Gradient-based attacks face several challenges. They solve a relaxation of the original combinatorial (discrete) optimization problem – the entries of the adjacency matrix are relaxed from $\{0, 1\}$ to $[0, 1]$. They need a differentiable proxy loss function since the actual objective of the attacker (e.g. accuracy) is often not differentiable, and the usual proxy such cross-entropy is suboptimal (Geisler et al., 2023). They assume white-box access to the model, including the structure and the weights. This limits the applicability or requires surrogate models. They can provide a false sense of security since defenses may be obfuscating gradients (Athalye et al., 2018; Geisler et al., 2023) and can get stuck in local minima. Their memory complexity grows quadratically w.r.t the number of nodes. Although the adjacency matrix is often sparse, the gradients w.r.t. it are not. As a result, tricks like block coordinate descent are needed (Geisler et al., 2021). We propose a model-agnostic evolutionary attack (EvA) that fixes all five of the above issues.

EvA explores the space of possible perturbations with a genetic algorithm (GA). Our approach operates in the discrete space of potential perturbations without information from gradients – avoiding relaxation. It directly optimizes the objective (like accuracy) as long as it provides a meaningful signal. In addition to eliminating the need for a differentiable proxy, this black-box access to the objective enables us to define a broader class of attacks. In fact, it allowed us to easily design two novel attacks on graphs that aim at decreasing the effectiveness of robustness certificates or that break conformal guarantees. EvA shows outstanding effectiveness on vanilla and adversarially trained models compared to SOTA attacks. Unlike the gradient-based attacks, our attack has a $\mathcal{O}(\epsilon \cdot E)$ memory complexity where $\epsilon$ is the perturbation budget, and $E$ is the number of edges. This is because instead of storing a squared block of gradients, which scales with the size of the adjacency matrix, we only store the edge perturbations as an index. During the evaluation, we also maintain the same sparsity in representation as the graph itself. To take advantage of the available free memory, we employ a batch evaluation approach that speeds up the optimization.

Adversarial attacks are supposed to be imperceptible. In images, this is modeled by a $L_p$-ball of a small radius. Similarly, for the graph structure, a commonly used metric is the $L_0$ ball, which allows changing of the node degree significantly. Since this may be perceptible, we can incorporate *constraints* in our attack that limit the number of perturbations per node, in addition to the global budget. Similar to gradient-based attacks (Geisler et al., 2021), we set this so-called local budget to a fraction of the node's original degree. Interestingly, in some cases, our constrained attack can even beat the best unconstrained gradient-based attack. Overall, EvA finds significantly better solutions compared to the previous state-of-the-art methods (Geisler et al., 2021; Gosch et al., 2024), which highlight the sub-optimality of gradient-based methods.

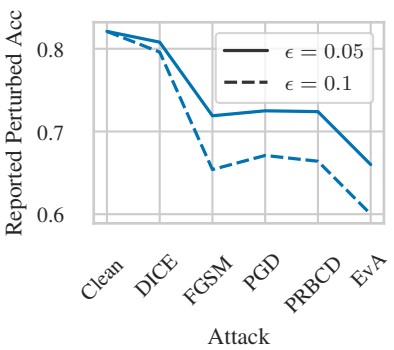

Figure 1: Reported performance of various attacks for transductive setting

Given the black-box nature of EvA, we were easily able to introduce the first graph certificate attack. One defense against adversarial attacks is to certify the prediction of a (smoothed) classifier (Bojchevski et al., 2020).

Certificates provide a robustness guarantee that the prediction will not change given a limited set of possible perturbations (e.g., at most $r_a$ additions and $r_d$ deletions). The certified ratio is the fraction of nodes for which the guarantee holds. Here, we define the attacker's objective as decreasing the certified ratio. EvA can decrease the ratio below the MLP level (which is by definition robust to any perturbation in structure), while also preserving the clean accuracy – making it less noticeable to a defender. We also introduce the first conformal attack on graphs. Conformal prediction (CP) converts any model's output to prediction sets with a guarantee to cover the true label with (adjustable) high probability. With EvA we can attack these conformal sets to either break the guarantee of increase the sets size (making them useless). While in principle one can design gradient-based attacks for these two new objectives, the amount of work is nontrivial since there are many non-differentiable components that would need to be relaxed. In contrast, for EvA, designing a new attack is simply a matter of changing the fitness function of the GA.

Importantly, perhaps the main contributions of this work is to highlight a scarcely explored research direction for attacks. Even off-the-shelf genetic algorithms significantly outperform gradient-based attacks. Fig. 1 compares EvA to the other attacks proposed over time. Our custom adaptive mutation further improves performance, but we argue that the space of evolutionary (and more broadly search-based) attacks has a lot of untapped potential.

## 2 BACKGROUND AND RELATED WORK

**Problem setup.** We focus on attacking the semi-supervised node classification task on graphs via perturbing a small number of edges. Formally, we are given a graph $\mathcal{G} = (\boldsymbol{X}, \boldsymbol{A}, \boldsymbol{y})$ in which $\boldsymbol{X}$ is the features matrix assigning a feature vector $\boldsymbol{x}_i$ to each node $v_i$ in the graph, $\boldsymbol{A}$ is the adjacency matrix (often sparse) that represents the set of edges $\mathcal{E}$, and $\boldsymbol{y}$ is the partially observable vector of labels. Nodes are partitioned into labeled and unlabeled sets $\mathcal{V} = \mathcal{V}_l \cup \mathcal{V}_u$. The GNN is trained on an observed subgraph $\mathcal{G}_{\mathrm{tr}}$ that includes the labeled nodes. Gosch et al. (2024) argue that the transductive setup, where $\mathcal{G}_{\mathrm{tr}} = \mathcal{G}$ is unrealistic since perfect robustness can be achieved by memorizing the training graph. Therefore, we mainly focus on the inductive setting where a model $f$ is trained on an induced subgraph $\mathcal{G}_{\mathrm{tr}} \subseteq \mathcal{G}$, validated on $\mathcal{G}_{\mathrm{val}} \subseteq \mathcal{G}$ and tested on $\mathcal{G}_{\mathrm{test}}$ where $\mathcal{G}_{\mathrm{tr}} \subset \mathcal{G}_{\mathrm{val}} \subset \mathcal{G}_{\mathrm{test}} = \mathcal{G}$.

**Threat model.** Our goal is to find a perturbation matrix $\boldsymbol{P} \in \{0, 1\}^{n \times n}$ that flips entities of the adjacency matrix $\tilde{\boldsymbol{A}} = \boldsymbol{A} \oplus \boldsymbol{P}$ to decrease the accuracy as much as possible. Here $n = |\mathcal{V}|$, and $\oplus$ is the element-wise XOR operator. For a given function $f$ as the GNN model, the accuracy is defined as $\sum_{v_i \in \mathcal{V}_{\mathrm{att}}} (1/|\mathcal{V}_{\mathrm{att}}|) \cdot \mathbf{1}[f(\mathcal{G})_{v_i} = y_i]$ where $\mathcal{V}_{\mathrm{att}}$ is the set of nodes that we attack. In global attacks this is usually the test nodes, while in targeted attacks the target is a single node. To keep the perturbations imperceptible, we assume that the adversary can only perturb up to $\delta := \epsilon \cdot |\mathcal{E}[\mathcal{V}_{\mathrm{att}} : \mathcal{V}]|$ edges where $\mathcal{E}[\mathcal{A} : \mathcal{B}]$ is the subset of edges between nodes in $\mathcal{A}$ and $\mathcal{B}$. Formally, for any generic

loss function $\mathcal{L}$,

$$\boldsymbol{P} = \arg\max_{\boldsymbol{P}} \quad \mathcal{L}(f(\mathcal{G}(\boldsymbol{X}, \boldsymbol{A} \oplus \boldsymbol{P}))_{\text{att}}, \boldsymbol{y}_{\text{att}})$$

$$s.t. \quad \boldsymbol{1}_N \boldsymbol{P} \boldsymbol{1}_N^\top \leq \epsilon \cdot |\mathcal{E}[\mathcal{V}_{\text{att}} : \mathcal{V}]| \tag{1}$$

Here $f(\cdot)_{\text{att}}$ returns the vector of predictions for the nodes in $\mathcal{V}_{\text{att}}$. In an evasion attack, $\mathcal{L}$ is the accuracy. Eq. 1 can include additional constraints like the local constraint from Gosch et al. (2023) that restrict the number of perturbation per node to some fraction (e.g., half) of its degree.

**Gradient-based attacks.** A common approach to attack the graph structure is to compute the gradient of the loss function w.r.t. the adjacency matrix. This requires a relaxation on the domain of $\boldsymbol{A}$ from $\{0, 1\}^{n \times n}$ to $[0, 1]^{n \times n}$. If the loss function is not differentiable (e.g., accuracy), then a differentiable surrogate like the categorical cross entropy or $\tanh$-margin (Geisler et al., 2023) is used instead. We compute the derivatives of the loss w.r.t. $\boldsymbol{A}$ and update the perturbation matrix. Finally, the edges are either sampled or rounded from the perturbation matrix, which returns the solution to the binary domain. There are various tricks to improve gradient-based attacks, but most follow a similar high-level procedure.

**Related Work.** Adversarial attacks on graphs are generally divided into two main categories: evasion attacks Xu et al. (2019); Zügner et al. (2018); Geisler et al. (2023); Gosch et al. (2024), where the attacker perturbs the graph after the model has been trained, and poisoning attacks Zügner et al. (2020); Lingam et al. (2023); Zügner et al. (2018), where the attacker modifies the graph prior to training. These attacks can be further classified into global attacks (e.g., Geisler et al. (2023); Zhu et al. (2023)), which target multiple node predictions simultaneously, and targeted attacks, which focus on a single node or a subset of nodes. The manipulations can involve altering node attributes, modifying edge structures, or introducing malicious nodes. The earliest adversarial attacks on graphs were inspired by techniques used on continuous data, utilizing gradients to approximate perturbations on inherently discrete edges Xu et al. (2019); Zügner et al. (2018); Geisler et al. (2023). Additionally, reinforcement learning has been employed as an alternative approach to execute adversarial attacks Dai et al. (2018). Attackers leverage reinforcement learning algorithms to refine their attack strategies and disrupt the learning process of GNNs Sun et al. (2023). Although some new attacks have been proposed in recent years (e.g., by Zhang et al. (2024; 2023); Wang et al. (2023)), they are all based on some traditional algorithms (like gradient-based methods).

## 3 EvA: EVOLUTIONARY ATTACK

Our evolutionary-based attack (EvA) uses a genetic algorithm (Holland, 1984) as a heuristic to directly optimize $Eq. 1$. We define an initial set of possible (candidate) perturbations – called "population" – and iteratively improve this population. In each iteration, the individuals in the population are ordered based on their fitness, specifically in terms of how much each individual decreases the accuracy. We draft the next population by keeping the best individuals and producing new ones as a function of them. Our population for the next iteration is finalized after a mutation which introduces additional randomness that helps with exploration. Each element in the population is a possible perturbation, which is encoded as a vector of indices where an edge is flipped.

**Genetic algorithm (GA) in EvA.** We can define a genetic solver through the definition of four main components. (i) Population: It is a set of feasible answers to the problem which gradually improve over iterations. In our case each population element is one potential perturbation on the adjacency matrix. We define mapping $\Pi : \boldsymbol{x}_{i,t} \in [\frac{n}{2}(n-1)]^\delta \mapsto [n]^2$ which is an enumeration on the upper triangle of the $n \times n$ adjacency matrix. With that, we define each candidate as set $\boldsymbol{s}_{i,t} \in [\frac{n}{2}(n-1)]^\delta$ which refers to a perturbation. The corresponding perturbation matrix $\boldsymbol{P}_{i,t}$ is simply defined as $\boldsymbol{P}_{i,t}[p, q] = \boldsymbol{P}_{i,t}[q, p] = 1 \Leftrightarrow \exists j : \boldsymbol{s}_{i,t}[j] = \Pi^{-1}(p, q)$ for $p < q$. (ii) Fitness: Is a notion of how close to optimal each population element is. Given any loss function $\mathcal{L}$ we define the fitness function fit : $[\frac{n}{2}(n-1)]^\delta \mapsto \mathbb{R}$, as $\text{fit}(\boldsymbol{s}) = \mathcal{L}(\boldsymbol{X}, \boldsymbol{A} \oplus \boldsymbol{P_s}, \boldsymbol{y})$. Note that this objective can be non-differentiable, such as accuracy. As long as the loss function has enough sensitivity to differentiate between various individuals, we use it directly as the fitness (see § 4 for extended discussion). (iii) Crossover: Is an operation that defines a new population element by combining two existing ones. The crossover operation at point $j$ defines a new candidate vector $\boldsymbol{s}_{\text{new}} = \text{cross}_j(\boldsymbol{s}_1, \boldsymbol{s}_2) := \boldsymbol{s}_1[: j] \bullet \boldsymbol{s}[j + 1 :]$ where $\bullet$ is the concatenation of two vectors. Crossover

operation with more than one point is defined recursively in the order of joints. The number of crossovers $k_{\text{cross}}$ is a hyperparameter (see § C), and their location is chosen randomly in the range of the perturbation size. (iv) Mutation: Is a random operation that allows further exploration. The function $\texttt{mutate} : [\frac{n}{2}(n-1)]^\delta \mapsto [\frac{n}{2}(n-1)]^\delta$ is a random mapping of a candidate to another. One simple mutation function changes each index with some mutation probability $p$ to some other index in the range (uniformly at random). In § 4 we discuss more advanced mutation strategies that significantly improve performance.

Given all the ingredients above, GA operates by iteratively evolving the population toward a good solution. The algorithm begins with an initial random population. In EvA, this population is a set of vectors $\mathcal{S}_0 = \{s_{i,0}\}_{i=1}^{n_p}$ with random elements, where $n_p$ is the number of candidates in the population. By definition, our candidates always encode a valid perturbation—the budget of the perturbation is enforced by the length of each candidate vector.

In each iteration, candidates are evaluated using the fitness function. Based on the fitness scores, an elite sub-population of parents and new children is selected to proceed to the next iteration, while the rest of the population is removed. To create a new child, parents are selected through a tournament selection process: in each tournament, $n_{\text{tour}}$ random parents are chosen, and the best among them is selected for crossover and subsequent mutation. This process repeats for $t$ generations.

**Sparse encoding of the attack.** The population in our framework is an encoding of the perturbation matrix. The naive way for encoding this problem is to create a boolean vector of size $N^2$ encoding which entries are flipped, which results in memory complexity of $\mathcal{O}(|\mathcal{S}|N^2)$ where $|\mathcal{S}|$ is the population size. Instead, we introduce an approach that leverages the sparse nature of the solution and reduces the complexity to $\mathcal{O}(|\mathcal{S}| \cdot \epsilon \cdot |\mathcal{E}[\mathcal{V}_{\text{att}} : \mathcal{V}]|)$. In this encoding, instead of retaining all possible edges, we only keep the indices of the edges we want to flip. Therefore, any element in the population $z \in \mathcal{S}$ is a vector of $p = \lfloor \epsilon \cdot |\mathcal{E}[\mathcal{V}_{\text{att}} : \mathcal{V}]| \rfloor$ dimensions where each entity of it is an index in adjacency matrix $z[i] \in \{1, \cdots n(n-1)/2\}$ with $n = |\mathcal{V}|$. We use diagonal enumeration of an upper triangular $n \times n$ matrix as the encoding (see § B). The perturbation vector can contain repeated elements. During the evaluation of the vector, we transform it to a perturbation matrix $P_z$, and we compute the perturbed adjacency $\tilde{A} = A \oplus P_z$. All the mentioned computations are in sparse representation, and each individual of the population takes $\mathcal{O}(\delta)$ space. Moreover, with this encoding, we directly enforce the global budget since the size of each individual in the population is by design the number of allowed perturbations.

**Acceptable fitness functions.** The fitness function in GA is accessed in a black-box manner. Therefore, properties like differentiability are not a requirement, which allows us to use the accuracy directly. However, for scenarios like targeted attacks, where the objective is to only misclassify a single node, the 0-1 loss function is not a suitable fitness function. In other words, with the 0-1 loss, random search and GA are practically equivalent. Ideally, small changes in the solution should be reflected in the fitness function as well. This sensitivity to various individuals prevents GA from remaining in local optima. In § 5, we discuss the choice of fitness in targeted attacks.

**Drawbacks.** The aforementioned setup is the very baseline variant of EvA. While already effective (outperforming SOTA), in Fig. 2, we resolve several drawbacks by changing the definition of the initial population and the mutation function. In the baseline variant, a population is allowed to contain perturbations that are outside of the receptive field of the GNN for $\mathcal{V}_{\text{att}}$. This means that (at least for initial generations) a proportion of the attacking budget is wasted on ineffective perturbations. Even for perturbations connecting nodes with both ends outside of $\mathcal{V}_{\text{att}}$, the defender can easily revert them by memorizing the training subgraph. In § 4, we discuss further improvements.

## 4 Enhancing the Search

In § 3 we defined the baseline evolutionary attack and discussed the possible drawbacks. As shown in Fig. 2 the baseline EvA already outperforms the SOTA. Additionally, with the following modifications we increase its effectiveness by a notable margin. The key insight is that the baseline attack, same as many gradient-based attacks defined their target space as the entire graph – entire space of $\frac{n}{2}(n-1)$ possible edges. As mentioned in § 2, perturbations that do have both endpoints in the training subgraph can be easily reverted just by memorizing the training subgraph. Additionally, perturbations outside of the receptive field of $\mathcal{V}_{\text{att}}$ are a waste of budget as they do not affect the prediction of the target nodes.

**Initial population.** Our baseline initial population consists of random perturbations in the entire space of $A$. This is a naive approach that disregards closeness to $\mathcal{V}_{\text{att}}$. Instead, we restrict the initial population to have at least one endpoint in $\mathcal{V}_{\text{att}}$. This can easily done by randomly sampling both endpoints, one inside $\mathcal{V}_{\text{att}}$ and one in $\mathcal{V}$, and then mapping the edges back to the indices via $\Pi$.

**Targeted and adaptive mutation.** After initialization, another way to balance the exploration and exploitation power of the algorithm is by introducing diversity in the population. In the baseline uniform mutation function, we change each edge to another random edge in $\mathcal{V}$ with some mutation probability $p$. Same as in initialization, we define the "targeted mutation (TM)" by restricting the new mutated edge to have at least one end-point in $\mathcal{V}_{\text{att}}$. Remarkably, this modification shows a significant improvement as shown in Fig. 2. Furthermore, when the attack succeeds in altering a node's prediction, additional perturbations connected to it do not gain any more performance. Therefore, we exclude them from the endpoint that was restricted to $\mathcal{V}_{\text{att}}$. Notably, we still allow those nodes to connect to other nodes in $\mathcal{V}_{\text{att}}$ as they can also increase the misclassification risk for other nodes. We call the latter approach "adaptive targeted mutation" (ATM).

**Stacking perturbations.** Each population (at each iteration of EvA) needs to evaluate every individual. This means that each individual requires a forward pass on the perturbed graph. As mentioned before, our population takes $\mathcal{O}(\delta)$ memory, and during the evaluation, we still maintain the sparse representation of the graph. Therefore, if the memory budget allows, we can evaluate several perturbations at once by combining perturbed graphs into one large (disconnected) graph and running only one forward pass. In practice, for small datasets like CoraML, we only run one forward pass per iteration, as the entire population of 1024 individuals can be evaluated once.

**Fitness Function.** As we mentioned in § 1, one of the problems with gradient-based methods is finding a differentiable proxy aligned with the main objective. To further understand the effect of the loss function on attacks, we conducted an additional experiment where we replaced the fitness function of EvA with the cross-entropy and margin-based loss functions, which have become popular in adversarial attacks as surrogates for accuracy. This experiment seeks to evaluate the effect of the fitness function on attack performance. The results,

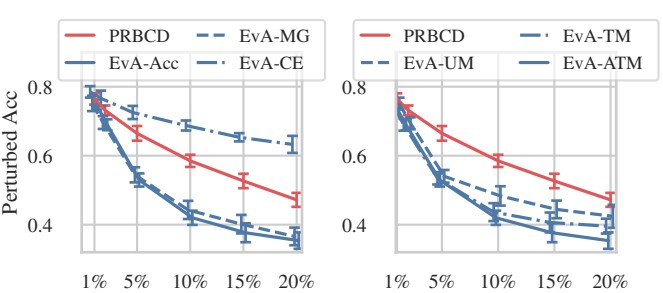

Figure 2: Effect of optimizing for different objective functions (left) and the influence of mutation type on EvA performance (right).

shown in Fig. 2, indicate that cross-entropy does not use the budget effectively. On the contrary, the margin-based loss provides a well-correlated surrogate loss. Since PRBCD also uses the margin-based loss, we see that the main reason for the large gap to EvA is not the loss function. We hypothesise that EvA, leveraging the exploratory capabilities of genetic algorithms, can more effectively explore the solution space and avoid bad local optima, while PRBCD gets stuck.

**Sensitivity.** The fitness landscape should be sensitive – small changes in the solution should ideally result in (at least some) changes in the fitness score. For EvA, higher sensitivity results in a better selection of the population for breeding and distinguishes even the smallest advantage of a specific individual. We empirically show that accuracy has enough sensitivity for the global attack and low to medium size budget. However, as we discuss in § 5 for targeted attacks, the variability of the fitness function decreases to two values $\{0, 1\}$. We discuss further in § D.1 why low sensitivity of the objective function makes GA-based methods close to random search.

**Effect of scaling.** A larger population provides greater diversity among solutions, which helps prevent early convergence to sub-optimal solutions, therefore the population size has a considerable impact on the performance of EvA. To observe this effect, we conducted experiments by changing the population size while keeping other parameters fixed on the PubMed dataset. For a fair comparison, we also attempted to scale PRBCD by increasing the number of steps and the size of the block

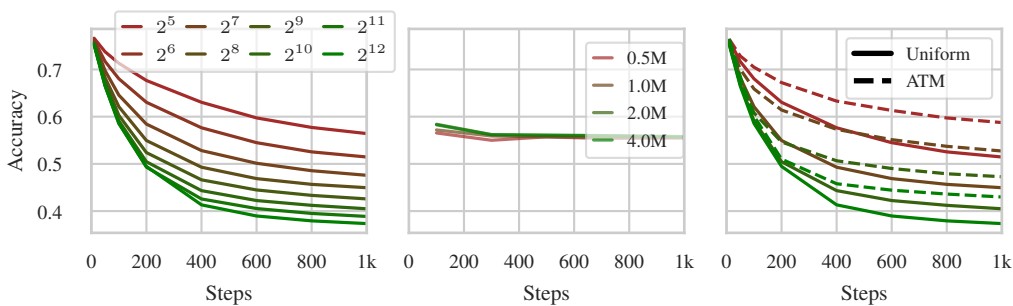

Figure 3: Effect of scaling on EvA and PRBCD performance (left, middle), the effect of mutation type and scaling on at $0.1\%$ budget (right) on Pubmed dataset.

coordinate subspace. In this experiment, we exponentially increased the block size, starting from 500 up to 4 million. As shown in Fig. 3 (left), increasing the population size improves EvA's ability to find better solutions by exploring the search space more effectively. Increasing the number of steps also increases the success rate of the attack. In contrast, PRBCD does not achieve further improvement by increasing the block size or the number of training steps.

We further investigate the effects of scaling and mutation types together. Fig. 3 (right) shows that adaptive targeted mutation can consistently enhance performance across all population sizes and outperform the uniform approach. This highlights the importance of selecting effective mutation strategies. Moreover, further exploration and refinement of mutation techniques could reveal more effective mutations, which could be explored in future studies.

## 5 OTHER OBJECTIVES

**Local attacks.** Gosch et al. (2024) argue that the perturbations within a global budget can still cause meaningful changes to the graph structure. For example, a perturbation might add edges to the node, increasing its degree to more than twice its current level while staying within the global budget. This can drastically alter the graph structure locally around that node, making the attack noticeable or, at the very least, impacting the graph's structural semantics. Therefore, they argue for a threat model that, in addition to a global budget, has a local budget that limits the number of perturbations per node to an $\epsilon_{\text{loc}}$ proportion of its degree.

**Local constrained mutation.** We enforce this constraint as a new mutation applied before finalizing the population. In our mutation, we count the row (or column) summation of the perturbation matrix which quantifies the number of edges added to (or removed from) each node. Calling the nodes with perturbation degree higher than $\epsilon_{\text{loc}} \cdot \deg(v_i)$ (the local perturbation budget) as "violating", we run an iterative refinement procedure where at each step we remove one edge from violating nodes and insert a non-violating edge instead. This refinement procedure continues until the local constraints are satisfied. Additionally, we rewrite the adaptive targeted mutation to account for the local budget – we restrict the mutation edges to those with remaining local budget for both end-points.

**Targeted attacks.** The targeted attack aims at one node to misclassify it with the least possible number of perturbations. With the discussion in § 4 the binary objective does not capture differences between different solutions. Therefore we use a proxy $\tanh$-margin loss as the fitness function.

### 5.1 ATTACKING NOVEL OBJECTIVES

In cases where the objective is not differentiable (e.g. accuracy), to apply gradient-based attacks, we need to find a differentiable surrogate that approximates the original objective. This is already discussed in § 4. Using these attacks becomes even more challenging when the attack objective is complicated and defined through several non-differentiable components (e.g., quantile computation or majority voting). Since our method nullifies the need for information from gradients, we can easily

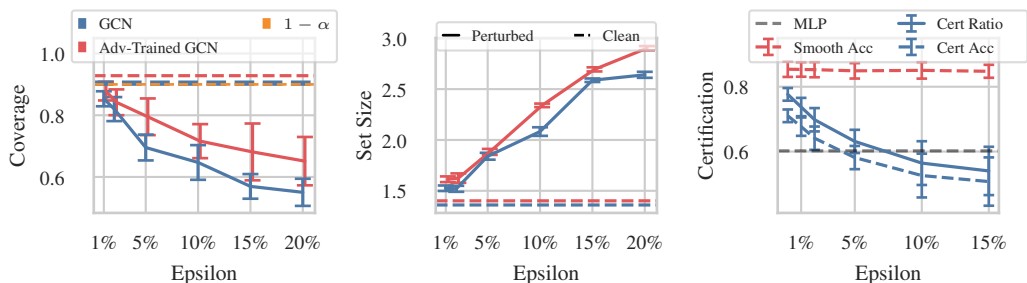

Figure 4: The conformal coverage (left) and conformal set size attack (middle) on Vanilla and adversarially trained GCN. The certificate attack (right) on GCN. All plots are for CoraML.

optimize for novel complex objectives. We define three new attacks on graphs: reducing the certified ratio of a smoothing-based model, decreasing the coverage, and the set size of conformal sets.

**Attacking randomized smoothing-based certificates.** A robustness certificate guarantees that the prediction of the classifier remains the same within the threat model. One way to obtain such a guarantee (for a black-box model) is through randomized smoothing. A smoothing scheme $\xi$ is a random function mapping an input $\boldsymbol{x}$ to a nearby point $\boldsymbol{x}'$ (e.g. additive isotropic Gaussian noise $\boldsymbol{x}' = \xi(\boldsymbol{x}) = \boldsymbol{x} + \boldsymbol{\epsilon}$, where $\boldsymbol{\epsilon} \sim \mathcal{N}(\boldsymbol{0}, \sigma^2 \boldsymbol{I})$). The convolution of the smoothing scheme and the classifier $\Pr[f(\boldsymbol{x} + \boldsymbol{\epsilon}) = y]$ changes slowly around $\boldsymbol{x}$ and this allows us to bound the worst-case minimum of the smooth prediction probability within $\mathcal{B}$. If this minimum is above $0.5$, we can certify that the smooth model returns the same label for any $\tilde{\boldsymbol{x}} \in \mathcal{B}(\boldsymbol{x})$. A possible adversarial objective is to reduce the number of nodes that are certified (a.k.a. certified ratio).

For certifying a prediction we compute the majority vote – the probability that the classifier predicts the top class for randomized $\boldsymbol{x}' \sim \xi(\boldsymbol{x})$. Then, we find a lower bound for this probability within the perturbation ball (see § D). Exact computation of the majority vote is generally intractable. Instead, we use Monte-Carlo (MC) sampling. These operations are not directly differentiable.

A naive implementation of the certified ratio objective is to compute $n_{\mathrm{mc}}$ random samples for each candidate perturbation $\tilde{\boldsymbol{A}}$. This makes the attack extremely slow as in each iteration, we need $n_p \cdot n_{\mathrm{mc}}$ samples, and for each Monte Carlo sample, we need $n^2$ samples from the Bernoulli distribution. Since statistical rigor is not crucial during the attack, we employ an efficient sampling strategy where we start with initial samples from clean $\boldsymbol{A}$, and for each perturbation, we only resample for the edges in $\tilde{\boldsymbol{A}} \triangle \boldsymbol{A}$. We use the stacked inference technique (see § 4) on MC samples which ultimately reduces the computation to one inference per each perturbation $\tilde{\boldsymbol{A}}$. Moreover, the certified radius is only a function of the smooth classifier's probability and it is non-decreasing w.r.t. it. This allows us to reduce all certificate computations to one binary search for the minimum required probability. Then the objective is to minimize the number of nodes with probability above this threshold. We further discuss this attack in § D.

**Attacking conformal prediction.** Instead of label prediction, conformal prediction (CP) returns prediction sets that are guaranteed to include the true label with $1 - \alpha$ probability. This post-hoc statistical method treats the model as a black-box and requires only a calibration set of labeled points whose labels were not used during model training. CP is applicable in both inductive and transductive Graph Neural Networks (GNNs) under the assumption of node-exchangeability (Zargarbashi & Bojchevski, 2024). Adversarial attacks on conformal prediction aim to decrease the empirical coverage by perturbing the input. In addition, we also define an attack that reduces the applicability of the prediction sets by increasing the average set size.

To compute prediction sets we need to compute a quantile from the set of true calibration conformity scores and compare the scores of the test node to the quantile threshold. This operation is again not directly differentiable which is not a problem for EvA. In our experimental setup, the defender calibrates on a random subset of $\mathcal{V}_u$ (besides the test, this is the only set with labels unseen by the model). Assuming that the unlabeled and test nodes are originally exchangeable (node-exchangeability), the conformal guarantee is valid in the inductive setup upon recalibration on the clean graph. By

perturbing the edge structure we can easily break this guarantee. Therefore our objective is to change the edge structure such that the coverage is minimized. Intuitively, this requires maximizing the distribution shift between the test and calibration scores. We can perform conformal prediction for each individual in the population, and we set the coverage of $\mathcal{V}_{\text{att}}$ as the objective function.

Since we don't know the exact subset of the unlabeled nodes taken as calibration, we can use the unlabeled set entirely as the calibration set. Given that the defender will randomly sample from unlabeled nodes during the calibration, the coverage remains roughly the same for exchangeable subsets of $\mathcal{V}_u$ (Berti & Rigo, 1997). To the best of our knowledge, so far this is the only adversarial attack on the graph structure to break conformal inductive GNNs. Similarly, by changing the objective to the negative average set size, we can easily attack the usability of prediction sets (see Fig. 4).

## 6 EMPIRICAL RESULTS

With our empirical evaluations (i) we show that current gradient-based attacks are still very far from optimal since EvA outperforms them by a notable margin. (ii) We show that EvA inherently results in attacks that perturb each node with less change in nodes' degree. This is even without posing local budget restrictions. (iii) We also show even by adding local restrictions EvA still outperforms other gradient-based local attacks. (iv) The effectiveness of EvA is consistent across various models, and vanilla or robust training setups. (v) With the black-box nature of the attack we introduce the first attack that reduces the certified ratio and the first attack that breaks conformal sets on graphs.

**Experimental setup.** We evaluate EvA on common graph datasets: Cora-ML (McCallum et al., 2004), Citeseer (Sen et al., 2008), and PubMed (Namata et al., 2012). Shchur et al. (2018) show that GNN evaluation is sensitive to the initial train/val/test split. Therefore, we averaged our results for each dataset/model over five different data splits. In contrast with common GNN attacks, Gosch et al. (2024) show that transductive setup carries a false sense of robustness. In other words, trivially one can gain perfect robustness just by memorizing the clean data; models with robust and self-training also show to exploit this flaw. Following them, we report our results in an inductive setting. We divide graph nodes into four subsets: training, validation, and testing, each with 10% of the nodes and we leave the remaining 60% as unlabeled data. For completeness, in § A we compare attacks in the transductive setting as well, where again EvA is more effective.

Following Lingam et al. (2023), we maintain the distribution of labels for sampling train, validation, and test nodes. This provides a more realistic scenario compared to commonly used methods, such as sampling for training and validation with the same count probability for each class. For completeness in § A we report various sampling setups. However the this does not change the order between methods. Further information about the model and hyperparameters can be found in § C.

**Attacking vanilla models.** As shown in Fig. 5, EvA outperforms the SOTA attack PRBCD by a significant margin. This comparison remains consistent across various datasets and models. We report these results extensively in § A. Interestingly, we show that in many vanilla and robust models, a very small budget $\epsilon \sim 0.05$ EvA drops the accuracy below the level of the MLP model. This is a condition where the model leveraging the structure works worse than a model that completely ignores edges.

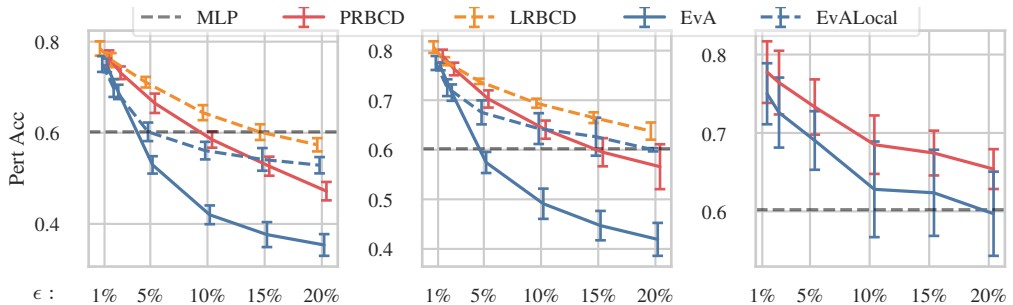

Figure 5: Performance of EvA on CoraML. From left to right the results are on Vanila GCN, adversarially trained GCN using PRBCD, and Soft-Median-GDC model.

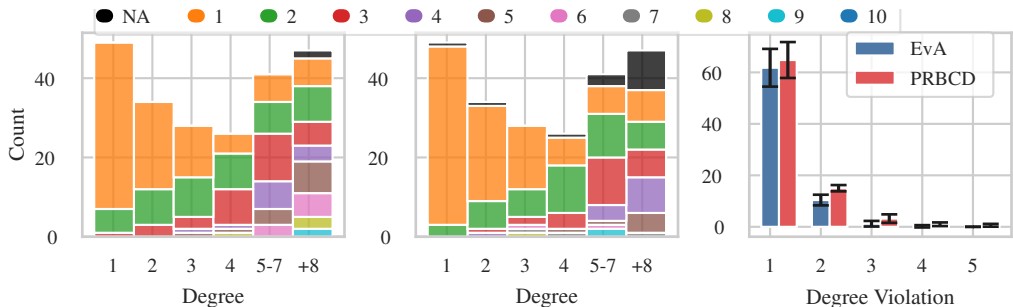

Figure 6: The number of perturbations (in different colors) that have been used by EvA (left) and PRBCD (middle) to target nodes with a specific degree. The right figures present the number of violations that EvA, PRBCD introduce (for $\epsilon_{\text{loc}} = 0.5$). NA (black) indicates a failed attack.

The SoftMedian model seems to show an inherent robustness to both EvA and PRBCD. Therefore, to break the model below the accuracy of MLP, we require $\geq 0.2$ perturbation budget. Even in the SoftMedian model, our attack is significantly more effective in comparison to PRBCD.

**Adversarially trained models.** Similar to vanilla models, EvA outperforms other approaches for robust models by a notable margin. As expected, models trained with EvA were shown to be more robust, and other attacks were less effective to them. However, this additional robustness is not significant. Table 9 (§ A) compares attacks in models with different adversarial training.

**Local attacks.** Building upon the discussion in § 5, we enforce the local constraint by adding a mutation function that iteratively removes edges exceeding the local budget. The local variant of EvA also shows to be consistently better than the local LRBCD attack. For the PubMed dataset EvA's effectiveness has a slower trend by increasing the budget $\epsilon$. On PubMed (see § A), when enforcing locally constrained mutation, the number of reconsidered edges increases due to the density of the graph. This makes the search significantly harder. On other datasets, though, EvA shows consistently better results and, more importantly, sharper decrease at lower $\epsilon$.

**Targeted attack.** We perform attacks on each node separately, with varying budgets from one to a maximum of 10 edges, until the prediction changes. As we discussed in § 5, the accuracy on one node is non-expressive. Therefore we use the $\tanh$-Margin proxy loss. Fig. 6 (left and middle) compares EvA and PRBCD in tagetted attack. Our results show that PRBCD performs better with a budget of one, but is outperformed by EvA for budgets of two and higher. For instance, on the CoraML dataset PRBCD fails to modify 16 nodes with a maximum of 10 changes (NA, black), whereas this number is reduced to only 2 nodes for EvA. This result is expected due to the combinatorial nature of the problem: for budgets up to two, a greedy approach can find the optimal solution, but as the budget increases beyond three, the problem becomes significantly more complex.

**Attacking certificates.** As shown in Fig. 4 (right) we reduce the certified ratio - the number of nodes that the certificate can guarantee for the specified threat model - to a ratio below the accuracy of the MLP model. The MLP model here is a baseline as it is robust to any structure perturbation by trivially ignoring edges. We report the ratio certified by sparse smoothing (Bojchevski et al., 2020) with $p_= 0.4$, and $p_+ = 2 \times 10^-5$. Here $p_+$, and $p-$ are Bernoulli parameters of flipping a zero or one. We reported the result for $\mathcal{B}_{0,3}$ which means 0 edge addition and 3 deletions. While we aim to decrease the certified ratio, a direct outcome is that the certified accuracy drops. For a 5% budget, the certified accuracy drops below MLP, which is resilient to any structural perturbation by definition. Notably, the clean (smooth) accuracy stays the same making this attack less noticeable. For this experiment we used the GPRGNN model with robust training using PRBCD attack.

**Attacking conformal prediction.** We report the first structure attack to inductive conformal GNN (Zargarbashi & Bojchevski, 2024). As shown in Fig. 4 (right) the coverage drops quickly as we increase the perturbation budget. As expected, in an adversarially trained model, we observe a slower decrease in the empirical coverage. Another interesting objective to attack is increasing the set size since it affects the usability of the prediction set. In Fig. 4 (middle) we show that both vanilla and robust models are vulnerable to this attack.

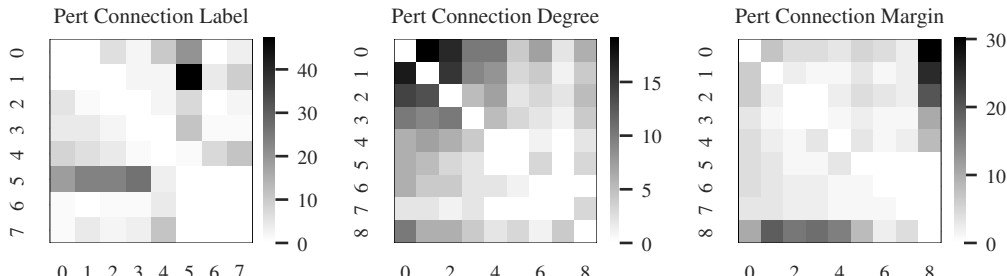

Figure 7: The upper triangle of each heatmap represents the perturbation connections for PRBCD, the lower triangle corresponds to the same for EvA, and the diagonal is set to zero.

**Local degree violation.** In this experiment, we did not enforce the local degree constraint in both EvA and PRBCD. However, we compared the final solutions to assess how often the solutions violated this constraint. Fig. 6 (right) shows that EvA generally produces more diverse attacks, utilizing the global budget more efficiently, which leads to improved performance. Overall, EvA violates less for any degree number and therefore in total.

**Label diversity.** We further conduct an ablation study on the solutions found by EvA and PRBCD under a specific budget of 10%. In this experiment, we keep all hyperparameters of EvA and PRBCD fixed and run them across 10 different seeds. We then compare the average solutions generated by each adversary. The left figure in Fig. 7 shows the number of connections across different labels. In both cases, the methods focus more on label 5 than on the others, but EvA distributes the connections more uniformly compared to PRBCD. The middle figure illustrates the nodes with original degrees ranging from 1 to greater than 8. The results indicate that, in both attacks, most of the budget is spent connecting to low-degree nodes. However, compared to PRBCD, EvA allocates more of the budget to higher-degree nodes. Additionally, we calculate the margin loss for each node in the original graph and discretize them into eight levels. As shown in the right figure of Fig. 7, EvA allocates more of the budget to higher-margin nodes, resulting in a non-trivial solution that achieves a better optimum. Finally, it seems that EvA identifies solutions that differ from greedy-based heuristic, which usually only targets low-degree or low-margin nodes.

## 7 CONCLUSION

In contrast to gradient-based adversarial attacks on graph structure, we developed a new attack (EvA) based on a heuristic genetic algorithm. By eliminating differentiation, we can directly optimize for the objective of the adversary (e.g. the model's accuracy). This black-box nature enables us to define complex adversarial goals, including attacks on robustness certificates and conformal prediction. Our novel attacks decrease the certified ratio, and conformal coverage, and increase the conformal set size. We propose an encoding that reduces the memory complexity of the attack to the same order as the perturbation budget which allows us to adapt to various computational constraints. Given the drastic decrease in the model's accuracy by applying EvA, we highlight that even SOTA gradient-based attacks are far from optimal. Our main message is that search-based attacks are underexplored yet powerful as shown by our results.

**Limitations.** We use an off-the-shelf genetic algorithm. Surely, there is room for designing search algorithms specific to the domain of the problem or hybrids of gradient and evolutionary search. As the graph size increases, the search space expands exponentially which makes convergence harder. While we remove the white-box assumption, we still assume the adversary has full knowledge of the graph and labels (same as most other attacks). This limitation can be easily addressed in future work. EvA uses many forward passes through the model which can be unrealistic in some attack scenarios. We leave the design of a further query-efficient variant for the future.

ETHICS STATEMENT

In this paper, we propose an adversarial attack without white-box access. However, our main focus is to point out the vulnerability of GNN models, opening the discussion on the need for more robust and reliable models, our results can be used to exploit the vulnerability of current GNNs.

REPRODUCABILITY

For reproducibility, we have uploaded the complete anonymized codebase on OpenReview.

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

# A SUPPLEMENTARY EXPERIMENTS

**Transductive Setting.** In § 6, we discussed that evaluating robustness in transductive setup is flawed since trivial robustness can be gained just by memorization of the clean graph (Gosch et al., 2024). However, for completeness, Table 1 reports the attacks' effectiveness in this setup. Consistent with other experiments, here also EvA outperforms SOTA. We provide the result for EvA, PRBCD, and for completeness, we also provide the result for PGA, which is a more recent attack.

Table 1: Classification accuracy (%) on the CoraML dataset in the transductive setting under different attacks and perturbation levels $\epsilon$. Results are averaged over multiple runs with standard deviations.

| Model | Attack | $\epsilon$ | | | | | |
|---|---|---|---|---|---|---|---|
| | | 0.01 | 0.02 | 0.05 | 0.10 | 0.15 | 0.20 |
| GCN | LRBCD | $80.0_{\pm 2.0}$ | $78.0_{\pm 2.0}$ | $73.0_{\pm 1.0}$ | $66.0_{\pm 1.0}$ | $61.0_{\pm 2.0}$ | $57.0_{\pm 2.0}$ |
| | PRBCD | $79.0_{\pm 2.0}$ | $77.0_{\pm 2.0}$ | $72.0_{\pm 2.0}$ | $65.0_{\pm 2.0}$ | $60.0_{\pm 2.0}$ | $56.0_{\pm 2.0}$ |
| | EvA | $77.0_{\pm 2.0}$ | $74.0_{\pm 2.0}$ | $66.0_{\pm 2.0}$ | $60.0_{\pm 2.0}$ | $59.0_{\pm 3.0}$ | $57.0_{\pm 3.0}$ |
| GPRGNN | LRBCD | $79.0_{\pm 3.0}$ | $77.0_{\pm 3.0}$ | $72.0_{\pm 4.0}$ | $63.0_{\pm 7.0}$ | $55.0_{\pm 12.0}$ | $49.0_{\pm 16.0}$ |
| | PRBCD | $79.0_{\pm 3.0}$ | $76.0_{\pm 4.0}$ | $70.0_{\pm 5.0}$ | $62.0_{\pm 7.0}$ | $55.0_{\pm 10.0}$ | $50.0_{\pm 13.0}$ |
| | EvA | $77.0_{\pm 3.0}$ | $73.0_{\pm 5.0}$ | $64.0_{\pm 6.0}$ | $57.0_{\pm 10.0}$ | $53.0_{\pm 13.0}$ | $50.0_{\pm 16.0}$ |

**Inductive Setting.** Here, we present additional results specifically for the inductive setting. Unlike the transductive setup, where robustness can be misleadingly achieved through memorization of the clean graph, the inductive framework provides a more comprehensive assessment of model performance in real-world scenarios. In this section, we detail the effectiveness of our method compared to other approaches.

Table 2: Classification accuracy (%) on the CoraML dataset in the inductive setting under different attacks and perturbation levels $\epsilon$. Results are averaged over multiple runs with standard deviations.

| Model | Attack | $\epsilon$ | | | | | |
|---|---|---|---|---|---|---|---|
| | | 0.01 | 0.02 | 0.05 | 0.10 | 0.15 | 0.20 |
| APPNP | EvA | $76.65_{\pm 1.32}$ | $71.03_{\pm 1.44}$ | $56.51_{\pm 1.60}$ | $49.32_{\pm 1.84}$ | $44.77_{\pm 2.04}$ | $41.42_{\pm 1.41}$ |
| | PRBCD | $78.65_{\pm 0.99}$ | $75.30_{\pm 1.27}$ | $68.75_{\pm 1.22}$ | $61.57_{\pm 1.65}$ | $55.44_{\pm 1.58}$ | $49.96_{\pm 2.42}$ |
| GAT | EvA | $64.20_{\pm 1.89}$ | $58.51_{\pm 2.45}$ | $40.99_{\pm 1.60}$ | $15.30_{\pm 4.47}$ | $9.40_{\pm 6.83}$ | $8.11_{\pm 6.65}$ |
| | PRBCD | $70.07_{\pm 2.82}$ | $66.55_{\pm 2.21}$ | $58.58_{\pm 3.33}$ | $49.61_{\pm 6.55}$ | $39.86_{\pm 6.78}$ | $36.94_{\pm 7.09}$ |
| GCN | EvA | $74.80_{\pm 1.50}$ | $68.97_{\pm 1.58}$ | $52.95_{\pm 1.91}$ | $41.99_{\pm 2.06}$ | $37.65_{\pm 2.74}$ | $35.37_{\pm 2.38}$ |
| | PRBCD | $76.44_{\pm 1.64}$ | $73.17_{\pm 1.39}$ | $66.48_{\pm 2.13}$ | $58.51_{\pm 1.77}$ | $52.67_{\pm 2.09}$ | $47.19_{\pm 2.02}$ |
| GPRGNN | EvA | $72.53_{\pm 4.11}$ | $66.83_{\pm 4.54}$ | $51.53_{\pm 5.57}$ | $42.21_{\pm 8.52}$ | $37.01_{\pm 9.83}$ | $34.52_{\pm 9.83}$ |
| | PRBCD | $74.95_{\pm 3.08}$ | $71.67_{\pm 2.76}$ | $64.84_{\pm 4.18}$ | $57.94_{\pm 4.55}$ | $53.24_{\pm 5.20}$ | $48.68_{\pm 6.52}$ |

**Stratified sampling.** Although unrealistic, in Table 8 we compare attacks in case the models are trained train/val/test sampled with the same number of nodes across different classes. Consistent with other results, EvA shows to be better here as well.

**Attacking accuracy of vanilla and robust models.** Table 9 compares EvA with SOTA PRBCD, and LRBCD. We compare both three attacks on vanilla models or models trained with adversarial examples of either of the attacks. Across all setups, EvA shows a comparably better performance.

## A.1 EXPERIMENTS

**Vanilla Models Experients.** For the experimental results, we mainly focus on the inductive setting introduced by (Lingam et al., 2023), where during training, we only use $\mathcal{G}_{\text{tr}}$, and during the attack, we target $\mathcal{G}_{\text{test}}$. We also provide results for the transductive setting, showcasing that our attack outperforms previous gradient-based methods, independent of the training setting. We conduct

Table 3: Classification accuracy (%) on the CoraML dataset in the inductive setting under different attacks and perturbation levels $\epsilon$. Results are averaged over multiple runs with standard deviations.

| Model | Attack | $\epsilon$ | | | | | |
|-------|--------|------|------|------|------|------|------|
| | | 0.01 | 0.02 | 0.05 | 0.10 | 0.15 | 0.20 |
| GCN | EvA | $74.80_{\pm1.50}$ | $68.97_{\pm1.58}$ | $52.95_{\pm1.91}$ | $41.99_{\pm2.06}$ | $37.65_{\pm2.74}$ | $35.37_{\pm2.38}$ |
| | EvaLocal | $75.09_{\pm1.73}$ | $69.82_{\pm1.96}$ | $60.21_{\pm2.04}$ | $56.09_{\pm1.93}$ | $54.16_{\pm2.48}$ | $52.88_{\pm1.79}$ |
| | LRBCD | $78.51_{\pm1.56}$ | $75.94_{\pm1.54}$ | $71.10_{\pm1.16}$ | $64.41_{\pm1.65}$ | $60.14_{\pm1.73}$ | $57.37_{\pm1.45}$ |
| | PRBCD | $76.44_{\pm1.64}$ | $73.17_{\pm1.39}$ | $66.48_{\pm2.13}$ | $58.51_{\pm1.77}$ | $52.67_{\pm2.09}$ | $47.19_{\pm2.02}$ |
| | PGA | $79.58_{\pm1.61}$ | $76.92_{\pm1.73}$ | $70.94_{\pm1.89}$ | $64.62_{\pm1.92}$ | $60.46_{\pm2.25}$ | $57.54_{\pm2.46}$ |
| GPRGNN | EvA | $72.53_{\pm4.11}$ | $66.83_{\pm4.54}$ | $51.53_{\pm5.57}$ | $42.21_{\pm8.52}$ | $37.01_{\pm9.83}$ | $34.52_{\pm9.83}$ |
| | EvaLocal | $73.31_{\pm3.30}$ | $67.26_{\pm4.17}$ | $58.29_{\pm7.96}$ | $53.38_{\pm11.42}$ | $51.10_{\pm12.66}$ | $49.96_{\pm13.63}$ |
| | LRBCD | $77.51_{\pm1.81}$ | $74.80_{\pm1.41}$ | $68.83_{\pm1.90}$ | $62.56_{\pm1.71}$ | $59.07_{\pm1.53}$ | $55.66_{\pm1.71}$ |
| | PRBCD | $74.95_{\pm3.08}$ | $71.67_{\pm2.76}$ | $64.84_{\pm4.18}$ | $57.94_{\pm4.55}$ | $53.24_{\pm5.20}$ | $48.68_{\pm6.52}$ |
| | PGA | $78.55_{\pm3.03}$ | $75.33_{\pm3.69}$ | $68.63_{\pm5.11}$ | $61.55_{\pm6.97}$ | $56.60_{\pm8.52}$ | $54.91_{\pm7.46}$ |

Table 4: Classification accuracy (%) on the Citeseer dataset in the inductive setting under different attacks and perturbation levels $\epsilon$. Results are averaged over multiple runs with standard deviations.

| Model | Attack | $\epsilon$ | | | | | |
|-------|--------|------|------|------|------|------|------|
| | | 0.01 | 0.02 | 0.05 | 0.10 | 0.15 | 0.20 |
| APPNP | EvA | - | - | $74.29_{\pm0.88}$ | $65.00_{\pm1.15}$ | $59.76_{\pm2.33}$ | $54.76_{\pm1.19}$ |
| | PRBCD | $87.26_{\pm0.90}$ | $85.48_{\pm1.49}$ | $81.79_{\pm1.08}$ | $76.55_{\pm0.68}$ | $72.44_{\pm1.66}$ | $69.29_{\pm1.69}$ |
| GAT | EvA | - | - | $67.14_{\pm3.65}$ | $51.19_{\pm4.21}$ | $37.74_{\pm4.94}$ | $27.62_{\pm9.49}$ |
| | PRBCD | $84.52_{\pm2.27}$ | $82.62_{\pm2.20}$ | $76.55_{\pm5.59}$ | $70.00_{\pm6.09}$ | $67.02_{\pm4.27}$ | $63.15_{\pm4.19}$ |
| GCN | EvA | $86.67_{\pm1.71}$ | $82.86_{\pm2.12}$ | $72.74_{\pm2.74}$ | $58.33_{\pm3.01}$ | $49.76_{\pm3.22}$ | $44.29_{\pm3.33}$ |
| | PRBCD | $87.38_{\pm1.81}$ | $85.83_{\pm2.43}$ | $80.95_{\pm2.06}$ | $74.29_{\pm4.22}$ | $69.76_{\pm4.34}$ | $67.62_{\pm4.96}$ |
| GPRGNN | EvA | $87.26_{\pm2.75}$ | $83.81_{\pm2.50}$ | $73.45_{\pm3.17}$ | $61.43_{\pm4.66}$ | $55.48_{\pm3.84}$ | $50.12_{\pm4.86}$ |
| | PRBCD | $88.45_{\pm2.29}$ | $86.31_{\pm2.45}$ | $82.02_{\pm2.61}$ | $77.14_{\pm2.84}$ | $73.93_{\pm3.89}$ | $69.64_{\pm3.47}$ |

experiments on the Cora-ML Citeseer, and Pubmed datasets, trained GCN, GPRGNN, APPNP and GAT. We run the attack for six different budgets (0.01, 0.02, 0.05, 0.1, 0.15, 0.2). Further details on training and attack hyperparameters are provided in C. We also use EvoTorch (Toklu et al., 2023) to impelement EvA.

**Robust Models.** Similarly, we provide the results for the adversarially trained model. In this case, during training, we use an adversarial attack at each step to attack $\mathcal{G}_{\text{tr}}$, and then we retrain the model on the adversarially perturbed graph $\tilde{\mathcal{G}}_{\text{tr}}$. The robust budget ($\epsilon_{robust}$) for adversarial attack during training was 0.02. This process repeats in each epoch of training until the model converges. We similarly use the inductive setting since, as Gosch et al. (2023) shows, in the transductive setup, the evaluation is flawed by a false sense of robustness. This originates from the fact that if, during the training process, the defender has access to perfect knowledge of all nodes in the graph, it can achieve perfect robustness by memorizing the clean structure of the graph in the model's weights. Table 9 presents our results in this setting. As the results indicate, EvA outperforms all previous attacks, even in adversarially trained models.

**Attacking Certificate.** As we discussed, EvA still be used in the case that we don't have access to a non-differentiable objective. In this experiment, we introduced a certificate attack which, to the best of our knowledge, is the first attack on certificates that reduces the certificate guarantees ....

Any possible appendices should be placed after bibliographies. If your paper has appendices, please submit the appendices together with the main body of the paper. There will be no separate supplementary material submission. The main text should be self-contained; reviewers are not obliged to look at the appendices when writing their review comments.

Table 5: Classification accuracy (%) on the Citeseer dataset in the inductive setting under different attacks and perturbation levels $\epsilon$. Results are averaged over multiple runs with standard deviations.

| Model | Attack | $\epsilon$ | | | | | |
|---|---|---|---|---|---|---|---|
| | | 0.01 | 0.02 | 0.05 | 0.10 | 0.15 | 0.20 |
| GCN | EvA | $86.67_{\pm1.71}$ | $82.86_{\pm2.12}$ | $72.74_{\pm2.74}$ | $58.33_{\pm3.01}$ | $49.76_{\pm3.22}$ | $44.29_{\pm3.33}$ |
| | EvaLocal | $87.38_{\pm1.65}$ | $83.57_{\pm2.17}$ | $78.21_{\pm3.17}$ | $76.43_{\pm2.62}$ | $75.00_{\pm3.23}$ | $74.52_{\pm3.16}$ |
| | LRBCD | $88.45_{\pm2.17}$ | $86.43_{\pm2.71}$ | $83.69_{\pm2.48}$ | $80.12_{\pm3.30}$ | $78.45_{\pm3.89}$ | $75.36_{\pm4.81}$ |
| | PRBCD | $87.38_{\pm1.81}$ | $85.83_{\pm2.43}$ | $80.95_{\pm2.06}$ | $74.29_{\pm4.22}$ | $69.76_{\pm4.34}$ | $67.62_{\pm4.96}$ |
| GPRGNN | EvA | $87.26_{\pm2.75}$ | $83.81_{\pm2.50}$ | $73.45_{\pm3.17}$ | $61.43_{\pm4.66}$ | $55.48_{\pm3.84}$ | $50.12_{\pm4.86}$ |
| | EvaLocal | $87.50_{\pm2.27}$ | $84.29_{\pm2.04}$ | $80.48_{\pm3.96}$ | $77.86_{\pm4.56}$ | $76.43_{\pm5.06}$ | $75.12_{\pm6.57}$ |
| | LRBCD | $89.76_{\pm2.50}$ | $87.98_{\pm2.48}$ | $85.12_{\pm2.76}$ | $81.90_{\pm2.83}$ | $79.64_{\pm4.08}$ | $78.45_{\pm4.92}$ |
| | PRBCD | $88.45_{\pm2.29}$ | $86.31_{\pm2.45}$ | $82.02_{\pm2.61}$ | $77.14_{\pm2.84}$ | $73.93_{\pm3.89}$ | $69.64_{\pm3.47}$ |

Table 6: Classification accuracy (%) on the PubMed dataset in the inductive setting under different attacks and perturbation levels $\epsilon$. Results are averaged over multiple runs with standard deviations.

| Model | Attack | $\epsilon$ | | | | | |
|---|---|---|---|---|---|---|---|
| | | 0.01 | 0.02 | 0.05 | 0.10 | 0.15 | 0.20 |
| APPNP | EvA | $73.85_{\pm2.35}$ | $69.64_{\pm2.16}$ | $57.07_{\pm2.32}$ | $47.03_{\pm2.18}$ | $43.94_{\pm1.83}$ | $41.93_{\pm2.18}$ |
| | PRBCD | $75.54_{\pm2.34}$ | $72.44_{\pm2.28}$ | $65.14_{\pm2.26}$ | $57.15_{\pm2.59}$ | $51.04_{\pm2.79}$ | $45.75_{\pm2.60}$ |
| GAT | EvA | $69.15_{\pm1.83}$ | $64.62_{\pm1.81}$ | $52.17_{\pm1.71}$ | $33.62_{\pm2.05}$ | $26.62_{\pm3.74}$ | $24.21_{\pm4.27}$ |
| | PRBCD | $71.33_{\pm1.53}$ | $67.78_{\pm1.79}$ | $59.73_{\pm2.10}$ | $49.87_{\pm1.36}$ | $42.04_{\pm1.57}$ | $35.94_{\pm1.66}$ |
| GCN | EvA | $72.60_{\pm2.19}$ | $68.35_{\pm2.41}$ | $56.15_{\pm1.92}$ | $42.93_{\pm2.64}$ | $40.46_{\pm2.76}$ | $39.11_{\pm2.98}$ |
| | PRBCD | $74.99_{\pm1.99}$ | $71.90_{\pm2.03}$ | $64.16_{\pm2.32}$ | $55.54_{\pm2.79}$ | $49.32_{\pm2.66}$ | $43.90_{\pm3.09}$ |
| GPRGNN | EvA | $72.01_{\pm4.18}$ | $67.61_{\pm4.28}$ | $55.95_{\pm4.32}$ | — | — | $42.39_{\pm9.63}$ |
| | PRBCD | $74.37_{\pm3.40}$ | $71.66_{\pm3.55}$ | $64.51_{\pm4.94}$ | $56.21_{\pm6.46}$ | $50.26_{\pm7.41}$ | $45.81_{\pm8.47}$ |

# B  TECHNICAL DETAILS OF EVA

**Mapping function: enumeration over $A$**  For enumerating over $A$, instead of using the row and column indices of the node to select, we introduced indexing. For a directed graph, the indexing starts from 0 to $n^2 - 1$. However, in an undirected graph, we only need the upper triangular part of the matrix $A$. To achieve this, we use the following algebraic solution to find the row and column of the perturbation by referencing only the upper triangular indexing.

$$
r = n - 2 - \left\lfloor \frac{\sqrt{-8l + 4n(n-1) - 7}}{2} - 0.5 \right\rfloor
$$
$$
c = 1 + l + r - \frac{n(n-1)}{2} + \left\lfloor \frac{(n-r)(n-r-1)}{2} \right\rfloor
$$
(2)

The advantage of this solution is that it can also be implemented in a vectorized way, making everything parallelizable.

# C  DATASETS AND MODELS, AND HYPERPARAMETERS

## C.1  STATISTICS OF DATASETS

In our experiments, we mainly conduct experiments on the commonly used graph datasets: Cora-ML, Citeseer, and PubMed, which are all representative academic citation networks. Their specific characteristics are as follows:

**Cora-ML.** The Cora-ML dataset contains 2,810 papers as nodes, with citation relationships between them as edges, resulting in 7,981 edges. Each paper is categorized into one of 7 classes corresponding

Table 7: Classification accuracy (%) on the PubMed dataset in the inductive setting under different attacks and perturbation levels $\epsilon$. Results are averaged over multiple runs with standard deviations.

| Model | Attack | $\epsilon$ | | | | | |
|---|---|---|---|---|---|---|---|
| | | 0.01 | 0.02 | 0.05 | 0.10 | 0.15 | 0.20 |
| GCN | EvA | $72.60_{\pm2.18}$ | $68.35_{\pm2.41}$ | $56.15_{\pm1.92}$ | $42.93_{\pm2.64}$ | $40.46_{\pm2.76}$ | $39.11_{\pm2.98}$ |
| | EvaLocal | $74.49_{\pm2.05}$ | $70.53_{\pm2.10}$ | $66.97_{\pm2.86}$ | $74.75_{\pm3.14}$ | $74.04_{\pm2.09}$ | $72.99_{\pm2.09}$ |
| | LRBCD | $74.89_{\pm2.04}$ | $71.48_{\pm2.49}$ | $65.68_{\pm2.90}$ | $60.24_{\pm3.15}$ | $56.81_{\pm3.02}$ | $54.07_{\pm2.99}$ |
| | PRBCD | $74.99_{\pm1.99}$ | $71.90_{\pm2.03}$ | $64.16_{\pm2.32}$ | $55.54_{\pm2.79}$ | $49.32_{\pm2.66}$ | $43.90_{\pm3.09}$ |
| GPRGNN | EvA | $72.01_{\pm4.18}$ | $67.61_{\pm4.28}$ | $55.95_{\pm4.32}$ | $-$ | $-$ | $42.39_{\pm9.63}$ |
| | EvaLocal | $73.36_{\pm3.71}$ | $69.68_{\pm3.93}$ | $65.61_{\pm6.75}$ | $70.64_{\pm2.85}$ | $72.27_{\pm5.06}$ | $71.40_{\pm5.41}$ |
| | LRBCD | $74.50_{\pm3.66}$ | $71.57_{\pm4.10}$ | $65.88_{\pm6.12}$ | $60.33_{\pm5.70}$ | $56.75_{\pm7.74}$ | $53.75_{\pm8.18}$ |
| | PRBCD | $74.37_{\pm3.40}$ | $71.66_{\pm3.55}$ | $64.51_{\pm4.94}$ | $56.21_{\pm6.46}$ | $50.26_{\pm7.41}$ | $45.81_{\pm8.47}$ |

Table 8: Classification accuracy (%) on the CoraML dataset in the inductive setting under different attacks and perturbation levels $\epsilon$. Results are averaged over multiple runs with standard deviations. Training, validation, and test sets are stratified.

| Model | Attack | $\epsilon$ | | | | | |
|---|---|---|---|---|---|---|---|
| | | 0.01 | 0.02 | 0.05 | 0.10 | 0.15 | 0.20 |
| GCN | LRBCD | $80.0_{\pm2.7}$ | $77.4_{\pm2.6}$ | $71.4_{\pm2.7}$ | $65.5_{\pm3.9}$ | $61.2_{\pm4.2}$ | $57.6_{\pm4.5}$ |
| | PRBCD | $78.7_{\pm2.8}$ | $75.3_{\pm3.4}$ | $67.9_{\pm3.4}$ | $59.5_{\pm3.5}$ | $53.0_{\pm4.1}$ | $48.4_{\pm3.7}$ |
| | EvA | $77.0_{\pm2.9}$ | $71.4_{\pm3.3}$ | $54.4_{\pm4.7}$ | $44.3_{\pm3.3}$ | $40.9_{\pm3.7}$ | $37.3_{\pm3.7}$ |
| GPRGNN | LRBCD | $76.2_{\pm7.9}$ | $73.1_{\pm7.9}$ | $66.1_{\pm11.5}$ | $60.5_{\pm11.7}$ | $56.4_{\pm12.4}$ | $53.4_{\pm12.9}$ |
| | PRBCD | $74.4_{\pm9.6}$ | $70.7_{\pm10.0}$ | $63.8_{\pm10.4}$ | $56.1_{\pm11.1}$ | $49.3_{\pm11.4}$ | $45.1_{\pm11.0}$ |
| | EvA | $71.9_{\pm10.4}$ | $65.0_{\pm11.4}$ | $50.1_{\pm11.4}$ | $41.8_{\pm14.0}$ | $37.2_{\pm14.6}$ | $35.2_{\pm15.7}$ |

to different subfields of machine learning. Each node is represented by a 1,433-dimensional bag-of-words (BoW) feature vector derived from the words in the titles and abstracts of the papers.

**Citeseer.** The CiteSeer dataset is also an academic citation network dataset consisting of 3,312 papers from 6 subfields of computer science and a total of 4,732 citation edges. Similar to Cora-ML, each paper as a node is represented by a BoW feature vector with a dimensionality of 3,703.

**PubMed.** The PubMed dataset is derived from a citation network of biomedical literature that contains 19,717 papers as nodes and 44,338 citation edges. Each paper is categorized into one of 3 classes based on its topic. The node features in PubMed are 500-dimensional vectors.

## C.2 DETAILS OF MODELS

In the following sections, we detail the hyperparameters and architectural details for the models performed in this paper. The experimental configuration files, including all hyperparameters, will be made publicly available upon acceptance of the paper.

**GCN.** We utilize a two-layer GCN with 64 hidden units. A dropout rate of 0.5 is applied during training.

**GAT.** Our GAT model consists of two layers with 64 hidden units and a single attention head. During training, we apply a dropout rate of 0.5 to the hidden units, but no dropout is applied to the neighborhood.

**APPNP.** We use a two-layer MLP with 64 hidden units to encode the node attributes. We then apply generalized graph diffusion, using a transition matrix and coefficients $\gamma_K = (1-\alpha)K$ and $\gamma_l = \alpha(1-\alpha)l$ for $l < K$.

**GPRGNN.** Similar to APPNP, we employ a two-layer MLP with 64 hidden units for the predictive part. We use the symmetric normalized adjacency matrix with self-loops as the transition matrix and

Table 9: Classification accuracy (%) on the CoraML dataset under different attacks and adversarial training methods. The results are averaged over multiple runs with standard deviations.

| Model | Adv. Tr. | Attack | $\epsilon$ | | | | | |
|---|---|---|---|---|---|---|---|---|
| | | | 0.01 | 0.02 | 0.05 | 0.10 | 0.15 | 0.20 |
| GCN | None | LRBCD | $78.51_{\pm1.56}$ | $75.94_{\pm1.54}$ | $71.10_{\pm1.16}$ | $64.41_{\pm1.65}$ | $60.14_{\pm1.73}$ | $57.37_{\pm1.45}$ |
| | | PRBCD | $76.44_{\pm1.64}$ | $73.17_{\pm1.39}$ | $66.48_{\pm2.13}$ | $58.51_{\pm1.77}$ | $52.67_{\pm2.09}$ | $47.19_{\pm2.02}$ |
| | | EvA | $74.80_{\pm1.50}$ | $68.97_{\pm1.58}$ | $52.95_{\pm1.91}$ | $41.99_{\pm2.06}$ | $37.65_{\pm2.74}$ | $35.37_{\pm2.38}$ |
| | LRBCD | LRBCD | $79.64_{\pm1.77}$ | $77.51_{\pm2.41}$ | $73.10_{\pm1.54}$ | $68.19_{\pm1.11}$ | $64.84_{\pm1.92}$ | $62.35_{\pm3.00}$ |
| | | PRBCD | $78.79_{\pm1.88}$ | $75.87_{\pm1.41}$ | $69.75_{\pm1.81}$ | $62.35_{\pm2.70}$ | $56.80_{\pm3.04}$ | $54.23_{\pm4.71}$ |
| | | EvA | $76.80_{\pm1.29}$ | $71.10_{\pm1.64}$ | $56.30_{\pm1.66}$ | $48.40_{\pm2.91}$ | $43.06_{\pm2.43}$ | $40.85_{\pm2.70}$ |
| | PRBCD | LRBCD | $80.71_{\pm1.16}$ | $77.86_{\pm0.81}$ | $73.81_{\pm0.54}$ | $69.40_{\pm0.91}$ | $66.48_{\pm1.08}$ | $63.77_{\pm1.73}$ |
| | | PRBCD | $78.93_{\pm1.27}$ | $76.30_{\pm1.27}$ | $70.25_{\pm1.74}$ | $64.06_{\pm1.83}$ | $59.50_{\pm2.84}$ | $56.58_{\pm4.53}$ |
| | | EvA | $76.80_{\pm0.77}$ | $71.53_{\pm1.65}$ | $57.44_{\pm2.13}$ | $49.11_{\pm3.05}$ | $44.70_{\pm2.96}$ | $41.92_{\pm3.32}$ |
| | EvA | LRBCD | $80.85_{\pm1.36}$ | $78.58_{\pm0.99}$ | $74.66_{\pm1.11}$ | $69.89_{\pm0.93}$ | $66.98_{\pm0.92}$ | $65.05_{\pm1.08}$ |
| | | PRBCD | $79.79_{\pm1.80}$ | $76.51_{\pm1.31}$ | $71.25_{\pm1.54}$ | $64.34_{\pm1.97}$ | $60.43_{\pm1.32}$ | $58.22_{\pm2.26}$ |
| | | EvA | $77.22_{\pm1.87}$ | $71.96_{\pm2.38}$ | $57.94_{\pm3.08}$ | $50.04_{\pm3.29}$ | $44.91_{\pm3.44}$ | $42.63_{\pm2.33}$ |
| GPRGNN | None | LRBCD | $77.51_{\pm2.81}$ | $74.80_{\pm3.08}$ | $68.83_{\pm4.20}$ | $62.56_{\pm4.69}$ | $59.07_{\pm5.98}$ | $55.66_{\pm6.99}$ |
| | | PRBCD | $74.95_{\pm3.08}$ | $71.67_{\pm2.76}$ | $64.84_{\pm4.18}$ | $57.94_{\pm4.55}$ | $53.24_{\pm5.20}$ | $48.68_{\pm6.52}$ |
| | | EvA | $72.53_{\pm4.11}$ | $66.83_{\pm4.54}$ | $51.53_{\pm5.57}$ | $42.21_{\pm8.52}$ | $37.01_{\pm9.84}$ | $34.52_{\pm9.83}$ |
| | LRBCD | LRBCD | $81.57_{\pm2.58}$ | $79.72_{\pm2.22}$ | $75.59_{\pm2.31}$ | $71.32_{\pm2.20}$ | $68.97_{\pm2.10}$ | $66.69_{\pm2.25}$ |
| | | PRBCD | $80.71_{\pm2.61}$ | $78.51_{\pm2.29}$ | $72.88_{\pm2.38}$ | $66.90_{\pm1.95}$ | $61.78_{\pm1.99}$ | $57.51_{\pm3.72}$ |
| | | EvA | $78.79_{\pm2.69}$ | $72.95_{\pm2.67}$ | $63.42_{\pm3.15}$ | $56.58_{\pm4.68}$ | $52.88_{\pm5.61}$ | $49.96_{\pm5.75}$ |
| | PRBCD | LRBCD | $80.43_{\pm2.01}$ | $78.01_{\pm1.91}$ | $73.74_{\pm1.66}$ | $69.96_{\pm2.14}$ | $67.19_{\pm2.51}$ | $64.84_{\pm3.20}$ |
| | | PRBCD | $80.21_{\pm2.43}$ | $77.30_{\pm2.63}$ | $71.53_{\pm2.67}$ | $65.12_{\pm3.21}$ | $60.07_{\pm4.10}$ | $55.37_{\pm3.85}$ |
| | | EvA | $78.79_{\pm2.45}$ | $73.10_{\pm2.54}$ | $62.85_{\pm4.93}$ | $56.94_{\pm6.64}$ | $53.74_{\pm7.65}$ | $51.60_{\pm8.10}$ |
| | EvA | LRBCD | $79.64_{\pm0.89}$ | $76.44_{\pm0.68}$ | $72.95_{\pm1.04}$ | $69.04_{\pm1.26}$ | $67.05_{\pm1.46}$ | $65.48_{\pm1.88}$ |
| | | PRBCD | $78.51_{\pm0.60}$ | $75.87_{\pm1.32}$ | $70.32_{\pm0.89}$ | $64.91_{\pm1.14}$ | $59.57_{\pm1.75}$ | $56.16_{\pm1.62}$ |
| | | EvA | $76.51_{\pm0.44}$ | $70.96_{\pm0.41}$ | $60.85_{\pm3.07}$ | $54.73_{\pm3.99}$ | $50.25_{\pm5.57}$ | $48.83_{\pm5.95}$ |

Table 10: Dataset Statistics

| Dataset | Nodes | Edges | Features | Classes |
|---|---|---|---|---|
| **Cora-ML** | 2,810 | 7,981 | 1,433 | 7 |
| **Citeseer** | 3,312 | 4,732 | 3,703 | 6 |
| **PubMed** | 19,717 | 44,338 | 500 | 3 |

randomly initialize the diffusion coefficients. We consider a total of $K = 10$ diffusion steps, with $\alpha$ set to 0.1. During training, we apply a dropout rate of 0.2 to the MLP, while no dropout is applied to the adjacency matrix. Unlike the method in Chien et al. (2021), we always learn the diffusion coefficients with weight decay, which acts as a regularization mechanism to prevent the coefficients from growing indefinitely.

**SoftMedian GDC.** We follow the default configuration from Geisler et al. (2023), using a temperature of $T = 0.2$ or the SoftMedian aggregation, with 64 hidden dimensions and a dropout rate of 0.5. We fix the Personalized PageRank diffusion coefficient to $\alpha = 0.15$ and apply a top $k = 64$ sparsification. During the attacks, the model remains fully differentiable, except for the sparsification of the propagation matrix.

**MLP.** We design the MLP following the prediction module of GPRGNN and APPNP, incorporating two layers with 64 hidden units. During training, we apply a dropout rate of 0.2 to the hidden layer.

### C.3 HYPERPARAMETER SETUP

In EvA we set the capacity of the computation to the same as the population, this means that all perturbations within a population are in one combined inference. However, in some cases where the

graph is large (e.g. PubMed), we reduce this number. Table 11 shows the hyper-parameter selection in almost all experiments. We only change the population number in some experiments, like certificate attacks, to reduce the computation. E.g., in the certificate attack, the population is reduced by a factor of 10.

### C.4 ATTACK HYPERPARAMETERS

To assess the robustness of GNNs, we utilize the following attacks and hyperparameters. Based on Geisler et al. (2023), we also select the tanh-margin loss as the attack objective.

**PRBCD.** We closely adhere to the setup outlined by Geisler et al. (2023). A block size of 500,000 is used with 500 training epochs. Afterward, the model state from the best epoch is restored, followed by 100 additional epochs with a decaying learning rate and no block resampling. Additionally, the learning rate is scaled according to $\delta$ and the block size, as recommended by Geisler et al. (2023).

**LRBCD.** The same block size of 500,000 is used with 500 training epochs. The learning rate is scaled based on $\delta$ and the block size, following the same approach as PRBCD. The local budget is consistently set as 0.5.

**EvA.** We set the population size to 1024 in most cases. Our mutation rate is 0.01, and increasing this number breaks the balance between exploration and exploitation, leading to less effective attacks. We run each attack for 500 iterations in most cases. In cases like certificate attacks, which are time-consuming, we reduce this number to 100. The details are summarized in Table 11.

**PGA.** For the PGA, we adopt the same setting as in Zhu et al. (2023). We use GCN as the surrogate model and tanhMarginMCE-0.5 as the loss type. The attack is configured with 1 greedy step, a pre-selection ratio of 0.1, and a selection ratio of 0.6. Additionally, the influence ratio is set to 0.8, with the selection policy based on node degree and margin.

Table 11: Hyper-parameters for PRBCD, LRBCD, and EvA

| Hyper-parameter | PRBCD | LRBCD | Hyper-parameter | EvA |
|---|---|---|---|---|
| Epochs | 500 | 500 | No. Steps | 500 |
| Fine-tune Epochs | 100 | 0 | Mutation Rate | 0.01 |
| Keep Heuristic | WeightOnly | WeightOnly | Tournament Size | 2 |
| Search Space Size | 500,000 | 500,000 | Population Size | 1,024 |
| Loss Type | tanhMargin | tanh-Margin | No. Crossovers | 30 |
| Early Stopping | N/A | False | Mutation Method | Adaptive |

## D  DETAILS ON NOVEL OBJECTIVES

**Smoothing-based certificate.** We define a randomized model as a convolution of the original model and a smoothing scheme. The smoothing scheme $\xi : \mathcal{X} \mapsto \mathcal{X}$ is a randomized function mapping the given input to a random nearby point. For graph structure, we use the sparse smoothing certificate (Bojchevski et al., 2020), which certifies whether within $\mathcal{B}_{r_a, r_d}$ the prediction of the smooth model remains the same. Here $r_a$ is the maximum number of possible additions, and $r_d$ is the maximum number of edge deletions. The smoothing function is defined by two Bernoulli parameters $p_+$, and $p_-$; i.e. for each entity of $\boldsymbol{A}$, if it is zero, it will be toggled with $p_+$ probability and otherwise with $p_-$.

The robustness certificate also accesses the model $f$ as a black box and defines a smooth model as $\bar{f}_y(\boldsymbol{x}) = \mathbb{E}[\mathbb{I}[f(\xi(\boldsymbol{x})) = y]]$ - each random sample $\boldsymbol{x}'$ is one vote for class $f(\boldsymbol{x}')$ and $\bar{f}_y$ is the proportion of votes for class $y$. Let $p = \bar{f}_y(\boldsymbol{x})$; the certificate finds a lower bound probability $\underline{p} \geq \min_{\tilde{\boldsymbol{x}} \in \mathcal{B}(\boldsymbol{x})} \bar{f}_y(\tilde{\boldsymbol{x}})$ and acts as a decision function $\mathbb{I}[\underline{y} \geq 0.5]$. In other words, the certificate returns yes, in case it is guaranteed that the smooth model will not return any value lower than 0.5 for class $y$ within $n\mathcal{B}(\boldsymbol{x})$. For further details about how to compute the certificate, see (Bojchevski et al., 2020).

**Adaptive sampling for certificate attack.** Statistical rigor is not a necessity while attacking the certificate. Therefore, while attacking, we can reduce the cost of resampling by only resampling the

subset of the graph that was perturbed. In other words, we initialize the search by computing samples $\boldsymbol{A}_1, \ldots, \boldsymbol{A}_m$, and for each perturbation $\tilde{\boldsymbol{A}}$ we only resample the edges in $\boldsymbol{A} \triangle \tilde{\boldsymbol{A}}$. For each edge in that set, if the edge was added via the perturbations, we resample $m$ Bernoulli variable with $p_-$, and otherwise $p_+$. We substitute those samples in the same entry of $\boldsymbol{A}_1, \ldots, \boldsymbol{A}_m$, and by running this process $|\delta|$ times, we assume that $\tilde{\boldsymbol{A}}_1, \cdots \tilde{\boldsymbol{A}}_m$ are representative as a new set of $m$ samples for $\tilde{\boldsymbol{A}}$. This adaptive sampling reduces the number of random computations from $m \cdot n^2$ to $m \cdot |\delta|$, which is significantly lower. Surely, to evaluate the final perturbation (the reported effectiveness), we don't use this approach, as it is statistically flawed and only applicable to reduce the computation during the attack.

## D.1 FITNESS FUNCTION

For a targeted attack, since the objective space is limited to the set $\{0, 1\}$, the sensitivity is very low. As a result, the method becomes equivalent to a random search. The zero-one fitness function means that all individuals with different perturbations receive the same score, causing the algorithm to behave more like a random search, as ties in each tournament are broken randomly. Secondly, since only one individual with a score of one is sufficient to halt the algorithm, all elite populations before success have scores of zero, which again results in random selection from them.

## D.2 PERFORMANCE ON ARXIV

To demonstrate the scalability of our attack on larger datasets, we also present results for the Arxiv dataset. For this, we consider two realistic scenarios. In the first scenario, similar to the previous one, the attacker has access to modify a limited number of edges. We provide results for three values of epsilon: $0.1\%, 0.5\%, 1.0\%$. Table 12 summarizes the results for this scenario. EvA outperforms PRBCD for smaller budgets and achieves comparable performance with a 1% budget, which could be further improved by scaling and increasing computational resources.

Table 12: Comparison of PRBCD and EvA performance for varying $\epsilon$ values

| Method | Clean | 0.1% | 0.5% | 1% |
|--------|-------|------|------|-----|
| PRBCD | 70.53 | 69.83 | 68.64 | 66.27 |
| EvA | 70.53 | 69.21 | 67.59 | 66.86 |

Alternatively, in a more practical scenario, the attacker compromise a subset of nodes (e.g., 1,000 nodes) and get access to them, referred to as control nodes, and strategically target a specific group within the network (the target group). For example, in a social network, an attacker could purchase 1,000 user accounts and use them to influence the performance of other subgroups. For this experiment, we randomly sampled 1,000 nodes five times and also randomly selected 1,500 nodes 5 time s as target group nodes. We then ran EvA and PRBCD and reported the average results in Table 13. Our method outperforms PRBCD in this scenario as well.

In summary, we demonstrate that our attack can be effectively applied and that it outperforms previous state-of-the-art methods.

Table 13: Comparison of PRBCD and EvA performance for varying $\epsilon$ values using contorl nodes

| Method | Clean | 1% | 5% |
|--------|-------|------|-------|
| PRBCD | | 64.89 | 54.7 |
| EvA | | 59.3 | 53.92 |

## D.3 TIME ANALYSIS

We also present an ablation study to compare the time analysis by evaluating the memory and wall clock time between EvA and the PRBCD method. In this experiment, we evaluate EvA with different numbers of steps, population sizes, and parallel evaluations, while PRBCD is run with varying

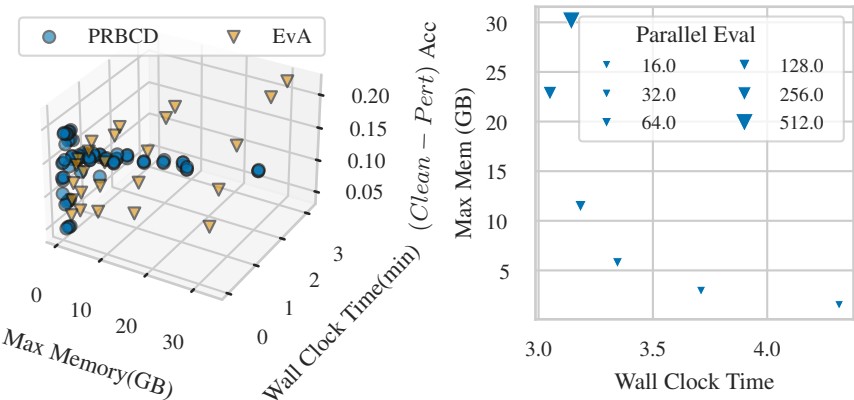

Figure 8: Comparing the memory usage between EvA and PRBCD

numbers of epochs and block sizes on the PubMed dataset. Fig. 8 (left) shows the results for EvA and PRBCD in terms of memory usage, wall clock time and method performance. Our method demonstrates comparable performance within the same level of wall clock time (less than a minute). Moreover, by increasing the wall clock time—through and memory either by a larger population size or more steps— EvA achieves additional benefits. Additionally, in Fig. 8 (right), we highlight how our framework provides a trade-off between time and memory for achieving the same level of accuracy by varying the number of parallel evaluations. For each point in the figure, we observe identical performance; however, the methods differ in memory usage due to different number of parallel evaluation, leading to variations in wall clock time.

### D.4 COMPARISON WITH (DAI ET AL., 2018)

(Dai et al., 2018) proposed a practical black-box attack (PBA), dividing it into PBA-C (with access to logits - continuous) and PBA-D (access only to the labels - discrete). As stated in (Dai et al., 2018), a genetic algorithm for global attacks requires PBA-C because it relies on logits, with the fitness function being the negative log-likelihood. We demonstrate that EvA not only eliminates the need for logits but also performs even better by directly optimizing for accuracy rather than using log-likelihood. To compare our method with (Dai et al., 2018), we modified the algorithm's fitness function and mutation mechanism to replicate the results reported in (Dai et al., 2018). This implementation retains scalability benefits, as it is also built upon our sparse encoded representation. Note here we re-implement Dai et al. (2018) in our sparse and parallelized framework. Their original implementation uses dense adjacency matrices and sequential evaluation and would achieve a significantly worse result within the same memory/run-time constraint. Even with our efficient re-implementation Dai et al. (2018) is significantly worse than ours. Table 14 provides the results for the `CoraML` dataset using the GCN architecture. EvA also significantly outperforms (Dai et al., 2018). Additionally, since our method is independent of gradients, we established the first attack on conformal prediction and certification. For conformal prediction, we attack coverage and set size where the latter criteria are not yet explored (to the best of our knowledge). Attacks tending to decrease certificate effectiveness are also under-explored in GNNs. In this work, we aim to achieve both attack on certified accuracy and certified ratio.

| Attack Name | Clean | 0.01 | 0.02 | 0.05 | 0.1 | 0.15 | 0.2 |
|---|---|---|---|---|---|---|---|
| (Dai et al., 2018) | $81.07_{\pm2.07}$ | $78.50_{\pm1.66}$ | $76.66_{\pm2.22}$ | $72.53_{\pm1.91}$ | $68.75_{\pm1.45}$ | $65.34_{\pm1.20}$ | $63.27_{\pm2.47}$ |
| EvA | | $81.07_{\pm2.07}$ $\mathbf{74.80}_{\pm1.50}$ | $\mathbf{68.97}_{\pm1.58}$ | $\mathbf{52.95}_{\pm1.91}$ | $\mathbf{41.99}_{\pm2.06}$ | $\mathbf{37.65}_{\pm2.74}$ | $\mathbf{35.37}_{\pm2.38}$ |

Table 14: Accuracy results of different attack methods under varying $\epsilon$ values.

