# OpenReview forum: "EvA: Evolutionary Attacks on Graphs"
_ICLR.cc/2025/Conference — Submitted to ICLR 2025_

### Official Review · Reviewer_DUCd · 2024-11-02

**Soundness:** 2
**Presentation:** 3
**Contribution:** 1
**Rating:** 5
**Confidence:** 4

**Summary:**

The work tackles the problem of adversarial attack for Graph Neural Networks (GNNs). The authors propose approaching the problem through  an evolutionary-based algorithm to solve the adversarial optimization problem without using a relaxation to the continuous space nor using gradient-based methods.

**Strengths:**

- The tackled problem of adversarial attacks is very interesting and proposing alternative the the widely used gradient-based method is important.
- Attacking both the adversarial certificates and specially the conformal guarantees extracted from the Conformal prediction framework is very novel and interesting.

**Weaknesses:**

- The usage of Genetic Algorithms in attacking adversarial attacks is not novel from my opinion. Specifically, previous work [1] have discussed the subject and proposed a method based on that. While the authors refer to this work in their related work section as proposing to tackle the subject from an RL perspective, they omit the fact that they also proposed a genetic-based attack denoted  GeneticAlg (Section 3.2.3 in their paper).
- I would also expect empirical results comparing your slightly different approach to their approach.


- In the experimental results, the authors evaluate only on the vanilla version of the models rather than considering adversarial defense methods. This latter therefore doesn’t confirm the effectiveness of the method as we are rather more interested in its performance when subject to the defense in comparison to the considered adversarial attacks.


- While the authors discussed the memory complexity of the methods, it seems that they didn’t provide any details about the time complexity and specially in comparison to the considered attacks.

—

[1] Adversarial Attack on Graph Structured Data. Dai & Al. - ICML 2018.

**Questions:**

- Would it be possible to provide more information about your data splits ? Why don’t you directly use the publicly available train/test splits (as they are provided for all the datasets you are considering)?
- Could you provide more details on the contrast of your proposed method to the method GeneticAlg [1] and also explain your claim about novelty of using Genetic Attacks.
- Additionally, would be important to provide the experimental results comapring your method to the GeneticAlg [1].
- Could you provide the results when subject to adversarial defenses ?
- Could you analyse the time complexity of your method in comparison to the considered attack baselines?

---

> ### Author Response · Authors · 2024-11-20
>
> **W1, W2, Q2, Q3.**  Compared to [1], we proposed three novel contributions: (i) Sparse and parallelized implementation (ii) Adaptive targeted mutation, and new fitness functions (Figure [2]) (iii) First ever attack on conformal prediction and GNN certificate.
>
> As we show in the table below [and table 14 in appendix], (i) and (ii) lead to significant improvements. More importantly, since 2018, genetic-based algorithms have been ignored in favor of the gradient-based attacks. We argue this is a big oversight of the field.
>
> Moreover, this perspective reveals the need to adjust new methods to balance exploration-exploitation for better performance. We believe this insight can suggest the design of new techniques that require fewer queries or hybrid approaches that combine the advantages of both strategies. For completeness we add further discussion in appendix.
>
> | **Attack Name** | **Clean** | **0.01** | **0.02** | **0.05** | **0.1** | **0.15** | **0.2** |
> | --- | --- | --- | --- | --- | --- | --- | --- |
> | GeneticAlg | 81.07 | 78.50 | 76.66 | 72.53 | 68.75 | 65.34 | 63.27 |
> | EvA | 81.07 | **74.80** | **68.97** | **52.95** | **41.99** | **37.65** | **35.37** |
>
> **W3, Q4.** We draw the attention of the reviewer to Fig.5 (center and left) and Table 9, where we report the result of EvA and SOTA on the two most effective and commonly used defenses: (i) adversarial training and (ii) robust message passing (e.g., SoftMedian [1]). We refer the reviewer to Fig. 5 (right) and Table 9 (appendix) for adversarial training. SoftMedian - established to be the strongest defense [5] - is also studied in Fig 5.
>
> **W4, Q5.** The reviewer is correct. We add the discussion on time complexity in the manuscript. The algorithm's runtime is O(population size/batch size * iterations * forward). The crossover and mutation do not have an effective overhead on the inference as they are linear to the size of the population. We added empirical results (fig 8 right) in appendix]on how EvA balances between time and memory constraints, i.e., with a limited memory budget, how much time is needed to attain the same performance. Furthermore, we showed the comparison with PRBCD with a limit, showing that for very small time limits (<20s), PRBCD is comparable to EvA. Increasing the computational capacity does not benefit PRBCD, but significantly benefits EvA.
>
> **Q1.** Following [2], we consider the inductive setup since the transductive setup is shown to carry a flawed sense of robustness. The public split are only for transductive setup. While, we report our results mainly on the (more realistic) inductive setup; Table 1 reports the transductive setup for completeness. As shown by [4], even in a transductive setup, evaluation on one split (e.g., the default) is a major pitfall. Therefore, the evaluation should be applied to several random splits. Consequently, we introduced a split using the pipeline outlined in [2]. Another critical problem with the original split is that the size of the validation set is much larger than training set which is both unrealistic and can lead to flawed evaluation as discussed at length in [3] (section 3, P1). Similarly, [1] provide the drawbacks of having a large validation set for semi-supervised learning in general (see Page 7 of their paper). To summarize, a more extensive validation set could be more effectively utilized in real-world applications as part of the training set.
>
> [1] Realistic Evaluation of Deep Semi-Supervised Learning Algorithms. Oliver et al., NeurIPS 2018.
>
> [2] Lukas Gosch, Simon Geisler, Daniel Sturm, Bertrand Charpentier, Daniel Zügner, and Stephan
>
> [3] Vijay Lingam, Mohammad Sadegh Akhondzadeh, and Aleksandar Bojchevski. Rethinking label poisoning for gnns: Pitfalls and attacks. In The Twelfth International Conference on Learning Representations, 2023.
>
> [4] Oleksandr Shchur, Maximilian Mumme, Aleksandar Bojchevski, and Stephan Günnemann. Pitfalls of graph neural network evaluation. arXiv preprint arXiv:1811.05868, 2018.
>
> [5] Mujkanovic, F., Geisler, S., Günnemann, S., & Bojchevski, A. (2022). Are defenses for graph neural networks robust?. *Advances in Neural Information Processing Systems*, *35*, 8954-8968.

---

> ### Author Response · Authors · 2024-11-25
>
> Thank you for your insightful feedback and time. If you have further questions, we’re happy to clarify. If your concerns are addressed, we’d appreciate it if you could consider updating your score.

---

### Official Review · Reviewer_XiUH · 2024-11-03

**Soundness:** 3
**Presentation:** 2
**Contribution:** 3
**Rating:** 8
**Confidence:** 5

**Summary:**

This paper proposes a new explore-and-exploit-based graph attack method. Unlike previous gradient-based methods, it no longer relies on approximate discrete optimization of the adjacency matrix. By eliminating the need to compute the gradients of a dense matrix, EVA significantly reduces the cost. Experimental results indicate a notable improvement compared to state-of-the-art methods.

**Strengths:**

1. The attack setting in this work is comprehensive, evaluating both local and global scenarios, and it selects a reasonably appropriate inductive setting.

2. Simple and effective with straightforward idea. Compared with gradient-based methods, e.g., PGD, PR-BCD, MetaAttack, EVA effectively reduce the cost.

**Weaknesses:**

## **Notations and mathematical expressions are unnecessarily complex**

EVA itself is a very simple method, and several improvements over the baseline-EVA (such as stopping the attack on nodes that are already misclassified) are not novel to this paper. However, the authors have chosen to use complex mathematical notation. Although this complexity does not hinder correctly understanding of the work, it creates unnecessary obstacles. Additionally, authors have chosen notations that is different from that used in this field, which I also believe is unnecessary. I think good writing should make it easier for readers to grasp the authors' ideas and methods, rather than trying to be distinctive in this regard.

## **Scalability**

The largest graph used in this work is PubMed. However, PRBCD can scale attack to ogb-arxiv, Products, and Papers 100M. Since one of the advantages of EVA is that it does not rely on computing gradients of the dense adjacency matrix, conducting experiments on larger-scale graphs would be more convincing.

**Questions:**

1. What is the time cost of EVA. Can authors provide some experiments to show this?

2. Can authors conduct experiments on larger graphs?

---

> ### Author Response · Authors · 2024-11-20
>
> We thank the reviewer for reading our paper and for the constructive feedbacks.  Below, we have provided a response addressing the comments and suggestions.
>
> **Larger graphs.** We conducted additional experiments on the larger OGBN-ArXiv dataset. These experiments are conducted in two setups which are designed to be realistic.
>
> **No limitation.** We apply the same experiment as for other datasets. We study the perturbation budget $\epsilon \in \{0.001, 0.005, 0.01\}$since even 1% equals approximately 10,000 edges on ArXiv. Higher perturbation budgets are unrealistic. As shown in the table, EvA performs better or comparable to PRBCD.
>
> | Epsilon | 0.1% (1084 edges) | 0.5% (5423 edges) | 1% (10847 edges) |
> | --- | --- | --- | --- |
> | PRBCD | 69.83 | 68.64 | 66.27 |
> | EvA | 69.21 | 67.59 | 66.86 |
>
> **Realistic larger scale attacks.** The freedom to choose any node and perturb its edges is unrealistic, especially in large networks. For instance, in a social network, an adversary might have a limited number of accounts that they control. Here, the adversary is allowed to perturb the edges with one endpoint landing in a predefined subset of “control” nodes. Moreover, often the attacker wants to target a subset of nodes, rather than all test nodes. To study we randomly pick 1000 control nodes, and 1500 target nodes. The table below shows the average result of over five random samplings. In this more realistic setting, our gap to PRBCD is even larger. Note, here the percentage is w.r.t. the edges of the control nodes.
>
> | Epsilon | 1%  | 5% |
> | --- | --- | --- |
> | PRBCD | 64.89 | 54.7 |
> | EvA | 59.3 | 53.92 |
>
> **Readability.** In light of the reviewer’s comment we improved the readability of the paper.
>
> **Time analysis.** We conducted a new experiment showing the performance of both EvA and PRBCD under various time and memory constraints. Both PRBCD and EvA offer adaptability through their hyper-parameters. In PRBCD, the block size can be adapted to accommodate memory limitations, and computations can be terminated within a specified time limit. Similarly, EvA can adjust by decreasing the population size and capping the number of iterations to meet time and memory restrictions. To explore these adaptations, we conducted an experiment (Fig. 8, left) that evaluates the wall-clock time and effectiveness of both approaches. The results indicate that PRBCD performs better under very tight time constraints (<20s), maintaining similar effectiveness even with increased computational power, which means that the method does not exhibit any improvement given more computational power.
> Furthermore, as shown in Fig. 3, similar accuracy can be achieved for various memory constraints by varying running time. For the PubMed graph, EvA surpasses PRBCD in under 1 minute, highlighting its superior scaling capabilities. The key takeaway from Fig. 3 is the distinct scalability of EvA, which is absent in PRBCD.

---

> > ### Comment · Reviewer_XiUH · 2024-11-20
> >
> > I have carefully reviewed the concerns and criticisms raised by other reviewers, and while they present valid points, I remain confident that this work makes a good contribution to this field. Therefore, I am happy to recommend its acceptance.
> >
> > I do have an additional problem regarding adaptive (white-box) attacks. Gradient-based attacks can effectively bypass certain heuristic defenses in graph neural networks by incorporating the differentiable components of these defense mechanisms [1]. Could the authors provide more details on how their method, specifically Eva, can conduct such adaptive attacks?
> >
> > [1] Are Defenses for Graph Neural Networks Robust?.

---

> > > ### Author Response · Authors · 2024-11-20
> > >
> > > Thank you for brining up this excellent point. Essentially EvA is adaptive "for free". Since it does not require gradient information it can handle even non-differentiable heuristic defences out of the box.  In fig. 5 (right), we conducted experiments on SoftMedian,  which is recognized as the strongest heuristic defense and demonstrated robustness against adaptive attacks [1]. Our results show that EvA outperforms the (adaptive) PRBCD, even in this scenarios.
> > >
> > > [1] Are Defenses for Graph Neural Networks Robust?

---

> > > > ### Comment · Reviewer_XiUH · 2024-11-21
> > > >
> > > > My initial concerns are basically addressed, so I increase my score to 8. However, I don't think EVA is essentially adaptive to any potential attacks. I think the definition of "adaptive" is that an attack can circumvent defenses by considering specific defense mechanisms. Incorporating gradient information of differentiable defense is just one possible way. If authors could provide some examples to show how EvA can be armed with adaptive designs agains some representative robust GNNS, this work can be more solid and comprehensive.

---

> > > > > ### Author Response · Authors · 2024-11-25
> > > > >
> > > > > We are happy that the concerns are addressed. Thank you for increasing the score.
> > > > > We will explore the idea of adaptive variants further.

---

### Official Review · Reviewer_DFpE · 2024-11-04

**Soundness:** 2
**Presentation:** 3
**Contribution:** 2
**Rating:** 5
**Confidence:** 4

**Summary:**

This paper introduces a new method for attacking Graph Neural Networks (GNNs) using evolutionary algorithms. Unlike traditional gradient-based attacks, which can be limited by the need for differentiable objectives and may yield suboptimal results, EvA directly tackles the discrete optimization problem of modifying graph structures. This approach enables EvA to operate in a black-box setting, without requiring gradient information, and can therefore work with any model or objective, including non-differentiable ones.

**Strengths:**

1. EvA presents an innovative approach to attacking GNNs through an evolutionary algorithm, moving beyond the limitations of traditional gradient-based methods.

2. The method introduces new attack objectives, including reducing robustness certificates and conformal prediction sets, marking the first exploration of these targets within graph adversarial attacks.

**Weaknesses:**

1. Even the authors have mentioned the mitigation of efficiency issues, the evolutionary search strategy can still be computationally expensive, especially for large graphs, and may struggle with convergence in high-dimensional search spaces.

2. Evolutionary algorithms are well-studied in adversarial attack scenarios outside of graph applications. The novelty of applying essentially similar algorithms within the graph setting remains a question.

3. The datasets in this paper, such as Cora-ML, Citeseer, and PubMed, are widely used benchmarks for evaluating GNN models, but they are relatively small in terms of the number of nodes and edges. This aligns with my first concern and can limit the evaluation's applicability to real-world scenarios where graphs, such as social networks, protein interaction networks, or large-scale citation networks, have millions of nodes and edges.

**Questions:**

My major concerns are the scalability and the novelty.

---

> ### Author Response · Authors · 2024-11-20
>
> We are thankful for the reviewer comment. Here we address the mentioned concerns.
>
> **W1, 3.** We conducted additional experiments on the larger OGBN-ArXiv dataset. These experiments are conducted in two setups which are designed to be realistic.
>
> **No limitation.** We apply the same experiment as for other datasets. We study the perturbation budget $\epsilon \in \{0.001, 0.005, 0.01\}$since even 1% equals approximately 10,000 edges on ArXiv. Higher perturbation budgets are unrealistic. As shown in the table, EvA performs better or comparable to PRBCD.
>
> | Epsilon | 0.1% (1084 edges) | 0.5% (5423 edges) | 1% (10847 edges) |
> | --- | --- | --- | --- |
> | PRBCD | 69.83 | 68.64 | 66.27 |
> | EvA | 69.21 | 67.59 | 66.86 |
>
> **Realistic larger scale attacks.** The freedom to choose any node and perturb its edges is unrealistic, especially in large networks. For instance, in a social network, an adversary might have a limited number of accounts that they control. Here, the adversary is allowed to perturb the edges with one endpoint landing in a predefined subset of “control” nodes. Moreover, often the attacker wants to target a subset of nodes, rather than all test nodes. To study we randomly pick 1000 control nodes, and 1500 target nodes. The table below shows the average result of over five random samplings. In this more realistic setting, our gap to PRBCD is even larger. Note, here the percentage is w.r.t. the edges of the control nodes.
>
> | Epsilon | 1%  | 5% |
> | --- | --- | --- |
> | PRBCD | 64.89 | 54.7 |
> | EvA | 59.3 | 53.92 |
>
> Additionally, we compared EvA and SOTA (PRBCD) in both memory and time in the modified version (Fig. 8), showing that (i) in the time-memory trade for various memory constraints, there is an instance of EvA with the same effectiveness. However, less available memory requires more time to achieve the same results. Furthermore, Fig. 8 (left) indicates that while PRBCD performs slightly better with very limited computation time (<20s), it fails to improve with increased computational resources. In contrast, EvA demonstrates scalable performance with additional computing. A similar experiment concerning memory constraint is shown in Fig. 3. Again, EvA scales with additional memory.
>
>
> **W2.** Compared to [1], we proposed three novel contributions: (i) Sparse and parallelized implementation (ii) Adaptive targeted mutation, and new fitness functions (Figure [2]) (iii) First ever attack on conformal prediction and GNN certificate.
>
> As we show in the table below [and table 14 in appendix], (i) and (ii) lead to significant improvements. More importantly, since 2018, genetic-based algorithms have been ignored in favour of the gradient-based attacks. We argue this is a big oversight of the field. Moreover, this perspective reveals the need to adjust new methods to balance exploration-exploitation for better performance. We believe this insight can suggest the design of new techniques that require fewer queries or hybrid approaches that combine the advantages of both strategies. For completeness we add further discussion in appendix.
>
> | **Attack Name** | **Clean** | **0.01** | **0.02** | **0.05** | **0.1** | **0.15** | **0.2** |
> | --- | --- | --- | --- | --- | --- | --- | --- |
> | GeneticAlg [1]  | 81.07 | 78.50 | 76.66 | 72.53 | 68.75 | 65.34 | 63.27 |
> | EvA | 81.07 | **74.80** | **68.97** | **52.95** | **41.99** | **37.65** | **35.37** |
>
>
> [1] Adversarial Attack on Graph Structured Data. Dai & Al. - ICML 2018.

---

> > ### Comment · Reviewer_DFpE · 2024-11-26
> > **Thanks for the response**
> >
> > Thank you for your response and clarifications. I appreciate your efforts and would like to provide further feedback.
> >
> > 1. While I acknowledge that there are some novelty of the proposed work, evolutionary and genetic attacks have been extensively explored in both graph and non-graph settings. Examples include graph-focused attacks such as GANI (Global Attacks on Graph Neural Networks via Imperceptible Node Injections), Unsupervised Euclidean Distance Attack on Network Embedding, and Lapa (Multi-Label-Flipping Adversarial Attacks on Graph Neural Networks). In non-graph domains, methods like Black-box Adversarial Attacks Using Evolution Strategies, Neural Networks Implemented by Differential Evolutionary Algorithms to Counter Attacks, CamoPatch (An Evolutionary Strategy for Generating Camouflaged Adversarial Patches), Art-Attack (Black-Box Adversarial Attack via Evolutionary Art), and EvoAttack (An Evolutionary Search-Based Adversarial Attack for Object Detection Models) provide robust baselines. Many of these non-graph methods are adaptable to graph settings. Most of them are not mentioned or compared in your work. Including and discussing these approaches would better contextualize the novelty of your method.
> >
> > 2. While introducing the attack on GNN conformal prediction is noteworthy, your work does not implement robust conformal prediction approaches with certified adversarial robustness, such as adversarially robust conformal prediction or provably robust conformal prediction with improved efficiency. These methods could provide valuable baselines or comparisons, strengthening the impact of your contributions in the realm of certifiable defenses and enhancing the practical relevance of your proposed attacks.
> >
> > 3. Regarding the other claimed contributions: Sparse and parallelized implementation is efficient, but sparsity-aware techniques and parallelized methods have been well-studied in the graph domain, which limits the degree of novelty here. Adaptive targeted mutation is an interesting addition, but adaptive strategies are not entirely new in the genetic algorithm literature.
> >
> > I have read the responses to other reviewers. In light of these points, I will keep my score.

---

> ### Author Response · Authors · 2024-11-25
>
> Thank you for your insightful feedback and time. If you have further questions, we’re happy to clarify. If your concerns are addressed, we’d appreciate it if you could consider updating your score.

---

> ### Author Response · Authors · 2024-12-02
>
> **Clarification on Related Work in Graph and Non-Graph Domains**
>
> GANI [1] uses genetic algorithm for node-injection attack, Lapa [2] explores label poisoning attacks and Unsupervised Euclidean Distance Attack [3] studies poisoning of unsupervised (random walk) embeddings. Since we focus on evasion under edge perturbation which is a different goal (in a few ways) from the above methods, they are all out-of scope as baselines, nonetheless, in the camera ready version we would surely discuss them in the related work section. We thank the reviewer for brining them to our attention.
>
> **Robust Conformal Prediction Approaches:**
>
> On robust conformal prediction [4] and [5] derive certified CP sets that apply for continuous data, but neither apply to graphs and binary data. [6] derives robust CP for graphs but their guarantees hold only in the transductive setting. Specifically, they assume that the defender trains, and computes the calibration scores on the **clean** graph. Perturbations on test nodes are then applied after calibration. However, as shown by [7] in this setup the defender can easily remember the clean graph and recover it after perturbation which results in full robustness for free.
>
> Deriving provably robust CP sets for inductive node classification is a non-trivial open problem. First of all, even in a non-adversarial setup, the coverage guarantee is only valid in the inductive setting after re-computing calibration nodes [8]. This is because the calibration nodes are affected during message passing with test nodes. Now, if those test nodes are potentially perturbed, the calibration nodes and their scores are also perturbed as a result. Therefore, we would need a certificate that accounts for both changes in calibration and test scores at the same time, which currently does not exist.
>
> **Contribution:**
>
> We want to highlight the comparison of EvA to the previous known work in genetic-based attack in the same setting [9]. While the high-level idea to use genetic algorithms is the same, compared to [9], EvA has several improvements (fitness, encoding, mutation, scalability) that make it roughly 2.5x more effective (e.g. 78 vs 74 for a budget of 1%) as shown in Table 14.
>
> Moreover, a similar argument on “novelty” can be made for gradient-based attacks. For example, both PGD and PRBCD have the same high-level idea of using project gradient descent. Yet, the details of how you instantiate a gradient-based attack are important for the final performance. This is why we have many different variants of gradient-based attacks accepted at top publishing venues (both for graphs and non-graph data).
>
> In any case, we show that we can get a huge gap to the previous SOTA attack (~11%), which we argue is a contribution enough of its own. Put differently, so far the field has largely ignored  genetic attacks (for which ever reason) and focused on gradient-based attacks. We argue that this is a mistake, and that the field should dedicate more effort to genetic-based or hybrid attacks. If this paper is not published, the field would likely continue on the same trend of gradient-based attacks which are far from optimal as we show.
>
>
> [1] Fang, J., Wen, H., Wu, J., Xuan, Q., Zheng, Z., & Chi, K. T. (2024). Gani: Global attacks on graph neural networks via imperceptible node injections. IEEE Transactions on Computational Social Systems.
>
> [2] Li, J., Li, H., He, J., & Dou, T. (2023, April). Lapa: Multi-label-flipping adversarial attacks on graph neural networks. In *2023 International Seminar on Computer Science and Engineering Technology (SCSET)* (pp. 29-32). IEEE.
>
> [3] Yu, S., Zheng, J., Chen, J., Xuan, Q., & Zhang, Q. (2020, July). Unsupervised euclidean distance attack on network embedding. In *2020 IEEE Fifth International Conference on Data Science in Cyberspace (DSC)* (pp. 71-77). IEEE.
>
> [4] Gendler, A., Weng, T. W., Daniel, L., & Romano, Y. (2021). Adversarially robust conformal prediction. In *International Conference on Learning Representations*.
>
> [5] Yan, G., Romano, Y., & Weng, T. W. (2024). Provably robust conformal prediction with improved efficiency. *arXiv preprint arXiv:2404.19651*.
>
> [6] Zargarbashi, S. H., Akhondzadeh, M. S., & Bojchevski, A. (2024). Robust yet efficient conformal prediction sets. *arXiv preprint arXiv:2407.09165*.
>
> [7] Gosch, L., Geisler, S., Sturm, D., Charpentier, B., Zügner, D., & Günnemann, S. (2024). Adversarial training for graph neural networks: Pitfalls, solutions, and new directions. *Advances in Neural Information Processing Systems*, *36*.
>
> [8] Zargarbashi, S. H., & Bojchevski, A. (2024). Conformal inductive graph neural networks. *arXiv preprint arXiv:2407.09173*
>
> [9] Dai, H., Li, H., Tian, T., Huang, X., Wang, L., Zhu, J., & Song, L. (2018, July). Adversarial attack on graph structured data. In *International conference on machine learning* (pp. 1115-1124). PMLR.

---

### Official Review · Reviewer_VB4Q · 2024-11-04

**Soundness:** 3
**Presentation:** 2
**Contribution:** 2
**Rating:** 5
**Confidence:** 4

**Summary:**

This paper introduces a new type of graph neural network (GNN) attack algorithm called Evolutionary Attack (EvA). EvA uses a genetic algorithm-based approach to directly optimize the graph structure in discrete space without the need for gradient information, making it adaptable to any black-box model and objective, including non-differentiable targets. By employing sparse encoding, EvA reduces memory complexity. Through adaptive mutation in genetic algorithms, it enhances higher performance. Experimental results show that EvA is more effective in reducing GNN accuracy than the SOTA methods, with an average accuracy decrease of about 11%. Furthermore, EvA has designed several new types of attacks and has shown high effectiveness in specific attack scenarios.

**Strengths:**

1. EvA does not rely on gradient information and can directly optimize in the discrete space. This innovative approach addresses some fundamental issues encountered by traditional gradient-based attack methods when dealing with graph-structured data, such as the need to relax discrete problems into continuous spaces and the reliance on non-differentiable objective functions.
2. The sparse encoding scheme significantly reduces the memory complexity of the attack, making it linearly related to the attack budget. This design not only improves the efficiency of the attack but also enables EvA to handle larger-scale graph data. Furthermore, by using batch evaluation methods, EvA can make more efficient use of computational resources, accelerating the optimization process, which is particularly important for large-scale graph data.
3. EvA does not require internal information of the model, which means it can be applied to black-box models, where we can only observe the input and output without access to the internal parameters. This feature greatly expands the scope of the attack method, enabling it to counter models that are commonly encountered in practice and protect their internal structure from being known to the outside world.

**Weaknesses:**

1.In the paper, it seems that the fitness function needs to be used frequently to analyze the population. The authors optimize the computation by using batched forward propagation and evaluating with the fitness function in one go. However, this also seems to require a significant amount of computational cost. It would be hoped that the authors conduct some experimental analysis comparing the time efficiency of EvA with SOTA attack methods.
2. In the experiments of this paper, the datasets used, such as CoraML, Citeseer, and PubMed, seem to be relatively small. When facing larger graphs, the computational cost of running the genetic algorithm will increase dramatically, and at this time, the performance of EvA may decline.
3. The layout of the article seems to have a few minor issues. For example, Fig. 2 is mentioned before Fig. 1 in the text.

**Questions:**

The paper proposes an effective attack method. However, I am curious if there are any specialized defense methods against this kind of genetic algorithm for adversarial attacks. If there are, it would make the work more comprehensive.

---

> ### Author Response · Authors · 2024-11-20
>
> We thank the reviewer for the comments. Here we address concerns:
>
> **W1.** Yes, indeed, EvA has more forward propagation compare to PRBCD. Through batched inference (highly parallelized), we reduce the time to even one forward pass per iteration for small graphs. Here, the batching process itself (definition of the large graph and extracting the evaluation for each tiny graph after inference) has no significant overhead while allowing hardware utilization. Despite the computational efficiency of PRBCD, it can only partially utilize the provided compute. Increasing the computation power also does not benefit PRBCD, while EvA benefits from it. Here, we conducted an experiment to precisely study this. In figure 8 in appendix, we show a 3D scatter plot where the 3 axes are time, memory and difference with clean accuracy. Under 20 sec and 5GB memory, PRBCD performs comparably with EvA. Increasing the runtime (from 1 to 4 minutes) and memory budget (1 to 30GB), EvA significantly improves while PRBCD’s performance remains the same. This result is on the relatively larger pubmed graph and is easily runnable in modern GPUs. The most important insight from Fig. 3 is precisely EvA's scaling property, which is absent in PRBCD.
>
> Besides, to address the time-space trade-off, Fig. 8 (right) shows the time-memory constraint runs, which all result in the same accuracy. This shows that EvA can adapt its runtime depending on the memory constraint.
>
> **W2.** We conducted additional experiments on the larger OGBN-ArXiv dataset. These experiments are conducted in two setups which are designed to be realistic.
>
> **No limitation.** We apply the same experiment as for other datasets. We study the perturbation budget $\epsilon \in \{0.001, 0.005, 0.01\}$since even 1% equals approximately 10,000 edges on ArXiv. Higher perturbation budgets are unrealistic. As shown in the table, EvA performs better or comparable to PRBCD.
>
> | Epsilon | 0.1% (1084 edges) | 0.5% (5423 edges) | 1% (10847 edges) |
> | --- | --- | --- | --- |
> | PRBCD | 69.83 | 68.64 | 66.27 |
> | EvA | 69.21 | 67.59 | 66.86 |
>
> **Realistic larger scale attacks.** The freedom to choose any node and perturb its edges is unrealistic, especially in large networks. For instance, in a social network, an adversary might have a limited number of accounts that they control. Here, the adversary is allowed to perturb the edges with one endpoint landing in a predefined subset of “control” nodes. Moreover, often the attacker wants to target a subset of nodes, rather than all test nodes. To study we randomly pick 1000 control nodes, and 1500 target nodes. The table below shows the average result of over five random samplings. In this more realistic setting, our gap to PRBCD is even larger. Note, here the percentage is w.r.t. the edges of the control nodes.
>
> | Epsilon | 1%  | 5% |
> | --- | --- | --- |
> | PRBCD | 64.89 | 54.7 |
> | EvA | 59.3 | 53.92 |
>
> **Q1.** There are two general approaches to making a GNN robust: (i) adversarial training and (ii) robust message passing (e.g., SoftMedian [1]). Both setups have already been studied in the paper. For adversarial training, we refer the reviewer to Fig. 4 (right) and Table 9 (appendix). SoftMedian - established to be the strongest defense [2] - is also studied in Fig 4.
>
> In light of the reviewer's comments, we have revised and updated the paper in OpenReview. In case any concerns are not fully addressed or are missing, we would be happy to discuss them further.
>
> [1] Geisler, S., Li, Y., Chen, J., Smola, A., & Loukas, A. (2021). *Robustness of Graph Neural Networks at Scale*. arXiv preprint arXiv:2110.14038.
>
> [2] Mujkanovic, F., Geisler, S., Günnemann, S., & Bojchevski, A. (2022). Are defenses for graph neural networks robust?. *Advances in Neural Information Processing Systems*, *35*, 8954-8968.

---

> ### Author Response · Authors · 2024-11-25
>
> Thank you for your insightful feedback and time. If you have further questions, we’re happy to clarify. If your concerns are addressed, we’d appreciate it if you could consider updating your score.

---

### Meta-Review · Area_Chair_Kybp · 2024-12-18

**Metareview:**

Post-discussion, this submission received mixed reviews: 5 (Reviewer VB4Q), 5 (Reviewer DFpE), 8 (Reviewer XiUH), and 5 (Reviewer DUCd). While Reviewer XiUH gave the highest rating, she/he did not actively advocate for acceptance during the discussion. The remaining reviewers maintained reservations, particularly regarding the novelty of the work, such as its comparison to existing evolutionary methods in adversarial settings. The paper's current treatment and positioning do not sufficiently address this concern. I recommend that the authors further revise the paper in line with the reviewers' comments to strengthen its novelty and positioning.

**Additional Comments On Reviewer Discussion:**

While Reviewer XiUH provided the highest rating, she/he did not champion the paper for acceptance during the discussion. The remaining reviewers maintained reservations, particularly regarding the work's novelty in comparison to existing evolutionary methods in adversarial settings.

---

### Decision · Program_Chairs · 2025-01-22

Reject